# The *Botrytis cinerea* Crh1 transglycosylase is a cytoplasmic effector triggering plant cell death and defense response

Kai Bi[1,2], Loredana Scalschi[1,3], Namrata Jaiswal[4], Tesfaye Mengiste[4], Renana Fried[1], Ana Belén Sanz[5], Javier Arroyo [5], Wenjun Zhu[2], Gal Masrati [6] & Amir Sharon [1✉]

Crh proteins catalyze crosslinking of chitin and glucan polymers in fungal cell walls. Here, we show that the BcCrh1 protein from the phytopathogenic fungus *Botrytis cinerea* acts as a cytoplasmic effector and elicitor of plant defense. BcCrh1 is localized in vacuoles and the endoplasmic reticulum during saprophytic growth. However, upon plant infection, the protein accumulates in infection cushions; it is then secreted to the apoplast and translocated into plant cells, where it induces cell death and defense responses. Two regions of 53 and 35 amino acids are sufficient for protein uptake and cell death induction, respectively. BcCrh1 mutant variants that are unable to dimerize lack transglycosylation activity, but are still able to induce plant cell death. Furthermore, *Arabidopsis* lines expressing the *bccrh1* gene exhibit reduced sensitivity to *B. cinerea*, suggesting a potential use of the BcCrh1 protein in plant immunization against this necrotrophic pathogen.

[1] School of Plant Sciences and Food Security, Faculty of Life Sciences, Tel Aviv University, Tel Aviv, Israel. [2] College of Life Science and Technology, Wuhan Polytechnic University, Wuhan City, Hubei Province, China. [3] Plant Physiology Area, Biochemistry and Biotechnology Group, Department CAMN, University Jaume I, Castellón, Spain. [4] Department of Botany and Plant Pathology, College of Agriculture, Purdue University, West Lafayette, IN, USA. [5] Dpto. Microbiología y Parasitología, Facultad de Farmacia, Universidad Complutense, IRYCIS, Madrid, Spain. [6] School of Neurobiology, Biochemistry and Biophysics, Faculty of Life Sciences, Tel Aviv University, Tel Aviv, Israel. ✉email: amirsh@tauex.tau.ac.il

*B*otrytis cinerea is a wide host-range necrotrophic plant pathogen that causes severe damages to crops worldwide[1,2]. The infection process includes an early stage, characterized by the formation of local necrotic lesions, an intermediate stage during which the lesions begin to spread at an increasing rate, and a late stage of constant lesion spreading[3]. A working model derived from these findings predicts secretion of cell death-inducing factors during the early stage that promote the formation of patches of dead tissue, which serve as foci for the subsequent stages[4]. This model has been supported by the discovery of secreted proteins with cell death-inducing activity, which are collectively referred to as necrosis-inducing proteins (NIPs)[5]. Broadly, all NIPs can be divided into two main classes: proteins that lack a known domain, and secreted enzymes that also induce plant cell death (henceforth referred to as catalytic NIPs).

The best-studied NIPs are a family of non-catalytic NIPs collectively named NEP or NELP/NLP (NEP-like proteins)[6,7]. Proteins in this family induce hypersensitive-like cell death in a variety of dicotyledonous, but not monocotyledonous plants[8–10]. Catalytic NIPs are usually glycosyl hydrolases (GHs) that degrade plant cell wall sugar polymers such as pectin[11], hemicellulose[5,12,13], and glucose polymers[14]. The GH NIPs that have been characterized so far remain in the apoplast after secretion by the fungus, and their cell death-inducing activity is mediated by plant extracellular membrane components, commonly in an SOBIR-BAK1-depedent manner[5,12,13]. Similar to the non-catalytic NIPs, hydrolase NIPs also induce plant defense, which in many cases was found to be unrelated to their catalytic activity[5,12–14]. In some instances, a short NIP-derived peptide fragment was found to be sufficient to induce necrosis and activate defense[13,15,16]. In other cases, disruption of the tertiary structure of the protein eliminated necrosis- and defense-inducing activities[5,17].

In search of novel NIPs, we have analyzed secretome that was collected from bean leaves after infection with *B. cinerea* spore suspension[5]. Here we report on the identification and characterization of BcCrh1, a GH16 transglycosylase that induces cell death and defense responses in plants. Crh (Congo red hypersensitivity) is a highly conserved family of proteins responsible for the cross-linking between chitin and glucan polymers in the fungal cell wall[18–20]. Hence, unlike other catalytic NIPs, BcCrh1 is not involved in plant cell wall degradation, but rather in fungal cell wall biosynthesis. Furthermore, we show that the BcCrh1 protein is internalized into the plant cell. This internalization is required for induction of cell death, meaning that BcCrh1 is a cytoplasmic effector, unlike most other NIPs that are apoplastic. We also found that BcCrh1 forms dimers, which seem to be necessary for the transglycosylase activity, while the monomeric protein is sufficient for induction of necrosis. Collectively, our study reveals an unexpected role for Crh proteins as mediators of fungal-plant interaction, and provides details on their role in fungal cell wall biosynthesis.

## Results

Proteomic analysis of *B. cinerea* secretome collected from infected leaves 28 hpi revealed the presence of 259 predicted proteins[5]. In search of cell death-inducing proteins, we cloned selected candidate genes and transiently expressed them in *Nicotiana benthamiana* leaves using *Agrobacterium tumefaciens*-mediated transformation (Agroinfiltration assay). Bcin01g06010, which induced strong cell death in *N. benthamiana*, was further characterized.

**Bcin01g06010 is a Crh family protein.** Sequence analysis of the Bcin01g06010 protein using SignalP-5.0 (http://www.cbs.dtu.dk/services/SignalP/) and TMHMM version 2.0 (http://www.cbs.dtu.

dk/services/TMHMM/) predicted a secretion signal at the N-terminal end and an absence of transmembrane helices, indicating that Bcin01g06010 is a secreted protein, in accordance with its presence in *B. cinerea* secretomes[5,21,22]. SMART (http://smart.embl.de/) analysis showed the presence of a conserved GH family 16 (GH16) domain between amino acid residues 49-263 (*E*-value 1.5e−36). A BLAST search revealed similarity to proteins in the Crh family (Supplementary Fig. 1), and 3D structure prediction showed strong structural similarity (probability 100%, *E*-value 1.1e−32) to *A. fumigatus* AfCrh5 (Supplementary Fig. 2a). The protein was named BcCrh1 based on the homology to the *Saccharomyces cerevisiae* Crh1 protein. A BLAST search of the NCBI database and published RNA sequencing data[23] revealed three additional *B. cinerea* Crh protein members that were named BcCrh2 (Bcin15g03070), BcCrh3 (Bcin13g03640) and BcCrh4 (Bcin07g04870). All four *B. cinerea* Crh proteins had a predicted secretion signal and a conserved GH16 domain (Supplementary Fig. 1a). Sequence alignment of the four *B. cinerea* Crh proteins with *S. cerevisiae*, *Candida albicans,* and *A. fumigatus* Crh homologues revealed the presence of the DEXDXE (enzymatic activity site) and the GTIXWXGG (the acceptor sugar binding site) motifs, which are highly conserved in all members of the Crh protein family (Supplementary Fig. 1b, c).

**BcCrh1 cell death-inducing activity is independent of its transglycosylase activity.** Agroinfiltration of *N. benthamiana* leaves with 35S:BcCrh1 triggered local cell death within five days post inoculation (Supplementary Fig. 3a). Infiltration assay with different concentrations of purified protein solution showed that 10 μM of BcCrh1 was sufficient to cause cell death in *N. benthamiana* and tomato leaves, but not in *A. thaliana*, and similar to other *B. cinerea* NIPs, BcCrh1 did not promote necrosis in monocots (Supplementary Fig. 4). To test whether the GH enzymatic activity of BcCrh1 was necessary for cell death-inducing activity, we mutated the conserved residues (MBcCrh1: E120Q/D122H/E124Q) at the putative catalytic site. Agroinfiltration or injection of leaves with the MBcCrh1 both produced similar necrotic spots as infiltration with the native BcCrh1 protein (Supplementary Figs. 3a, 4, 5), indicating that induction of plant cell death is unrelated to the enzymatic activity of BcCrh1.

To verify the hypothetical transglycosylase activity of BcCrh1, we conducted a complementation assay of a *S. cerevisiae* crh1Δ crh2Δ double mutant strain. This strain, lacking any chitin-glucan transglycosylase activity, is hypersensitive to Congo red. Complementation of this mutant with the *bccrh1* gene fully restored the wild-type phenotype (Supplementary Fig. 6). In contrast, yeast strains expressing *bccrh1* with mutations in the putative catalytic residues (MBcCrh1) retained the Congo red hypersensitive phenotype. These results confirmed that BcCrh1 has a transglycosylase activity similar to the *S. cerevisiae* Crh1 and Crh2 proteins and that catalytic residues E120, D122, and E124 are essential for this activity.

**BcCrh1 is a cytoplasmic effector.** To determine the site of interaction with the plant, we performed an Agroinfiltration assay of *N. benthamiana* leaves. Two types of constructs were used, for expression of the native and enzymatic inactive proteins with (SP(PR3)-BcCrh1[121–391] and SP(PR3)-MBcCrh1[121–391]) and without (BcCrh1[121–391] and MBcCrh1[121–391]) the *A. thaliana* pathogenesis-related protein 3 (PR3)[5] secretion signal.

The first and second sets of constructs result in the secretion of the protein outside the plant cell, or intracellular protein accumulation, respectively. Transfection of *N. benthamiana* leaves with Agrobacterium expressing any of the four constructs

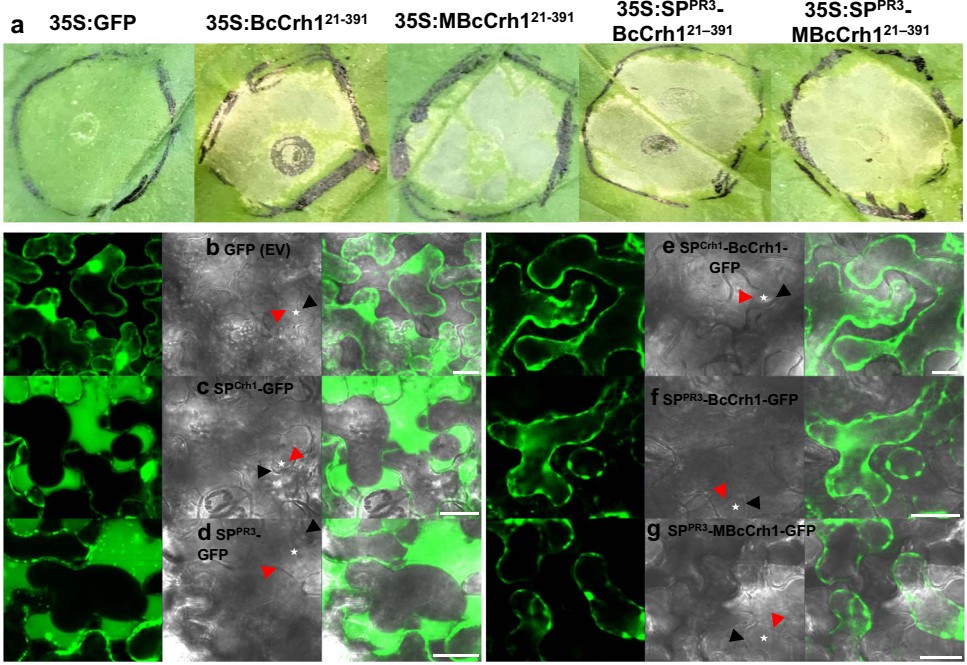

**Fig. 1 Localization of BcCrh1 inside plant cells is required for induction of cell death. a** Plants were infiltrated with Agrobacterium strains that were transformed with the *bccrh1* gene with or without secretion signal (SP). Images of necrotic lesions (**a**) were taken five days after Agroinfiltration. 35S:GFP: free GFP (control); 35S:BcCrh1$^{-SP}$ and 35S:MBcCrh1$^{-SP}$: native and enzyme inactive BcCrh1 respectively, without secretion signal; 35S:SP$^{PR3}$-BcCrh1$^{21-391}$ and 35S:SP$^{PR3}$-MBcCrh1$^{21-391}$: fusion of the native and enzyme inactive BcCrh1 respectively, with PR3 plant signal peptide; **b–g** Subcellular localization of GFP-fusion proteins. Leaves were harvested two days after Agroinfiltration, submerged for 20 min in 0.8 M mannitol to induce plasmolysis, and then samples were scanned by a confocal microscope. White asterisks mark apoplastic space between the cell wall (black arrow) and plasma membrane (red arrow) in plasmolysed plant cells. Left panel shows images of cells following Agroinfiltration with free GFP (control), right panel shows images following Agroinfiltration with BcCrh1-GFP fusion protein. **b** GFP without SP; **c** GFP fused to the BcCrh1 SP; **d** GFP fused to PR3 SP; **e** BcCrh1-GFP with native SP; **f** BcCrh1-GFP fused to PR3 SP; **g** MBcCrh1-GFP fused to PR3 SP. Bars = 20 μm. All the above experiments were repeated at least three times with similar results.

induced similar cell death levels (Fig. 1a, Supplementary Fig. 5), suggesting that cytoplasmic localized BcCrh1 or MBcCrh1 induce plant cell death. To further investigate the localization of secreted BcCrh1 in plant tissues, we infiltrated plant leaves with Agrobacterium expressing a BcCrh1/MBcCrh1-GFP fusion protein with a secretion signal. Confocal fluorescence microscopy showed that the fusion protein accumulated inside the plant cells, in contrast to control leaves that were treated with Agrobacterium expressing free GFP in which the fluorescent signal was observed exclusively in the apoplastic space (Fig. 1b–g). These results confirmed that the localization of BcCrh1 is inside plant cells.

**A 35-residue region of BcCrh1 is sufficient for the plant cell death-inducing activity.** To define the minimal region required for the cell death-inducing activity of BcCrh1, we generated a series of N-terminal and C-terminal deletions and tested their cell death-inducing activity. Long deletions at the C' end, including BcCrh$^{11-284}$, BcCrh$^{11-214}$, and BcCrh$^{11-144}$, retained full activity. However, when residues 75–284 were deleted, with (BcCrh$^{11-74}$) or without (BcCrh$^{121-74}$) the secretion signal, cell death-inducing activity was lost (Fig. 2a, b). These results indicated that residues 75–144 are necessary for cell death induction. Transient expression of BcCrh1$^{75-144}$, but not BcCrh1$^{145-391}$, induced cell death in *N. benthamiana* leaves, confirming that the region between amino acids 75–144 is both necessary and sufficient for induction of plant cell death. When BcCrh1$^{75-144}$ was fused to the PR3 secretion signal (SP$^{PR3}$-BcCrh1$^{75-144}$), the cell death activity was lost (Fig. 2a, b). Confocal images of *N. benthamiana* cells transiently expressing BcCrh1$^{75-144}$ or SP$^{PR3}$-BcCrh1$^{75-144}$ GFP

fusions, confirmed intracellular and apoplastic localization of the proteins, respectively (Fig. 2c). Following further analyses of different parts of the protein, the region necessary for induction of cell death was narrowed down to 35aa between residues 93–127 (Fig. 2a, b). Since BcCrh1$^{93-127}$ contains the catalytic pocket, we mutated the catalytic residues (E120Q/D122H/E124Q) in BcCrh1$^{93-127}$. The mutated epitope (MBcCrh1$^{93-127}$) still triggered cell death (Fig. 2b), verifying the earlier results obtained with MBcCrh1.

**A 53-residue region mediates uptake of BcCrh1 by plant cells.** In search of specific parts of BcCrh1 that might mediate protein uptake by plant cells, we performed Agroinfiltration assays of *N. benthamiana* leaves with constructs containing the secretion sequence and various segments of the protein. The expectation was that the protein segments containing the uptake signal would first be secreted out of the plant cell and then induce cell death following processing of the secretion signal and uptake by the plant cells (as in Fig. 1). No necrosis was expected by segments lacking the uptake signal. These analyses revealed that deletion of amino acids 21–74 (BcCrh1$^{21-74}$) prevented the induction of necrosis (Fig. 2a, b), suggesting that the first 53 aa following the secretion signal mediate uptake of the BcCrh1 protein by plant cells. To verify this result, we fused the BcCrh1$^{1-74}$ (includes the secretion signal and the putative uptake signal) to the apoplastic NIP BcXYG1[5]. Agroinfiltration with this construct failed to induce cell death, unlike control treatment with the native BcXYG1, which induced strong necrosis at 3 dpi (Fig. 3a). Consistent with the Agroinfiltration result, infiltration of tomato

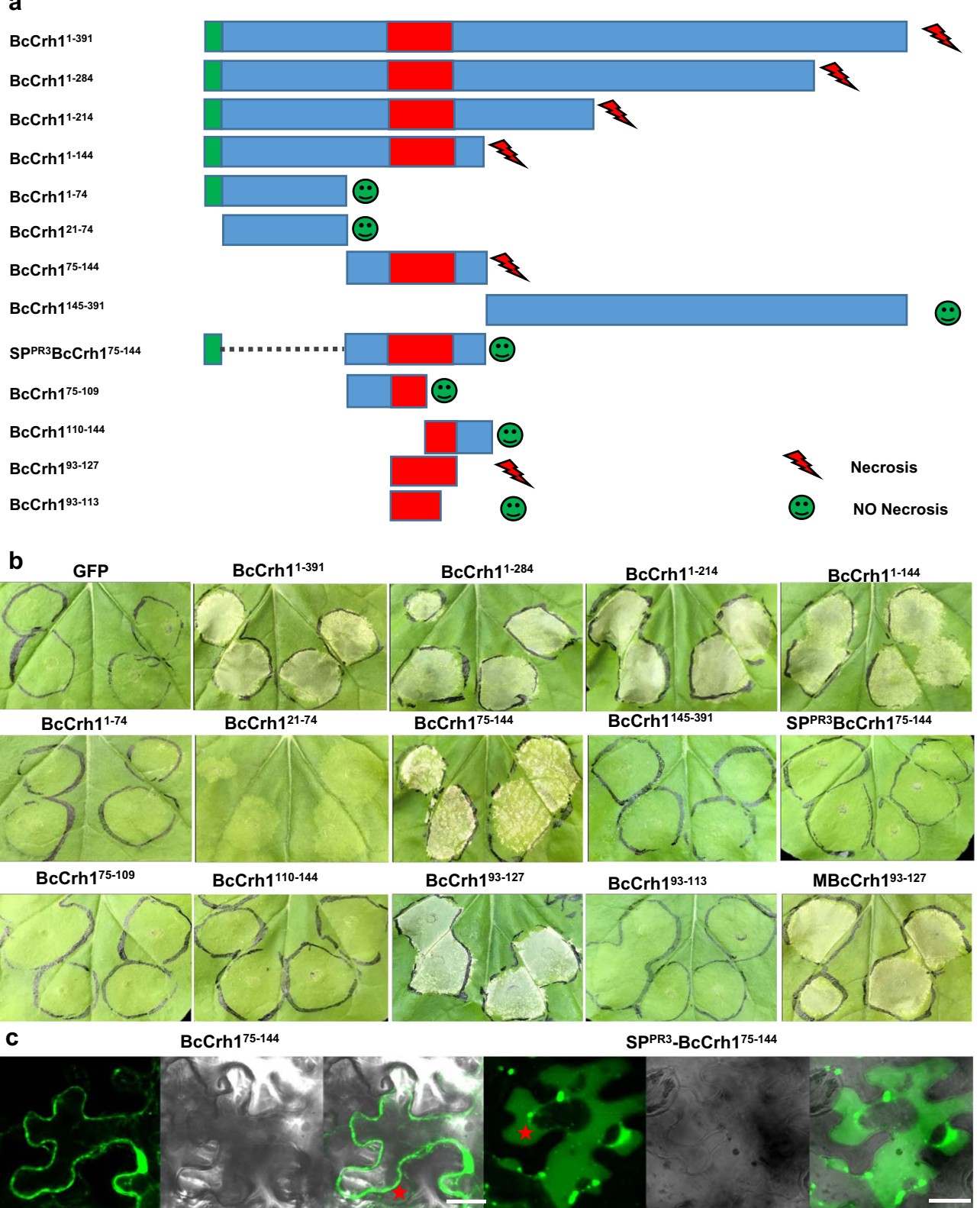

**Fig. 2 A stretch of 35 amino acids of BcCrh1 is necessary and sufficient for the induction of plant cell death. a** Schematic presentation of different BcCrh1 fragments that were used in Agroinfiltration assays. Green—SP, Red—the minimal region that was found to be sufficient for induction of cell death, Blue—other regions of the tested fragment. **b-c** representative images of leaves following infiltrations with Agrobacterium strains that contain constructs encoding the different BcCrh1 fragments. **b** Pictures were taken five days after Agroinfiltration; **c** Leaves were harvested two days after Agroinfiltration, submerged for 20 min in 0.8 M mannitol to induce plasmolysis, and then samples were scanned by a confocal microscope. Typical apoplastic space between the cell wall and plasma membrane in plasmolysed plant cells is marked with red asterisks. Bars = 20 μm. All the above experiments were repeated at least three times with similar results.

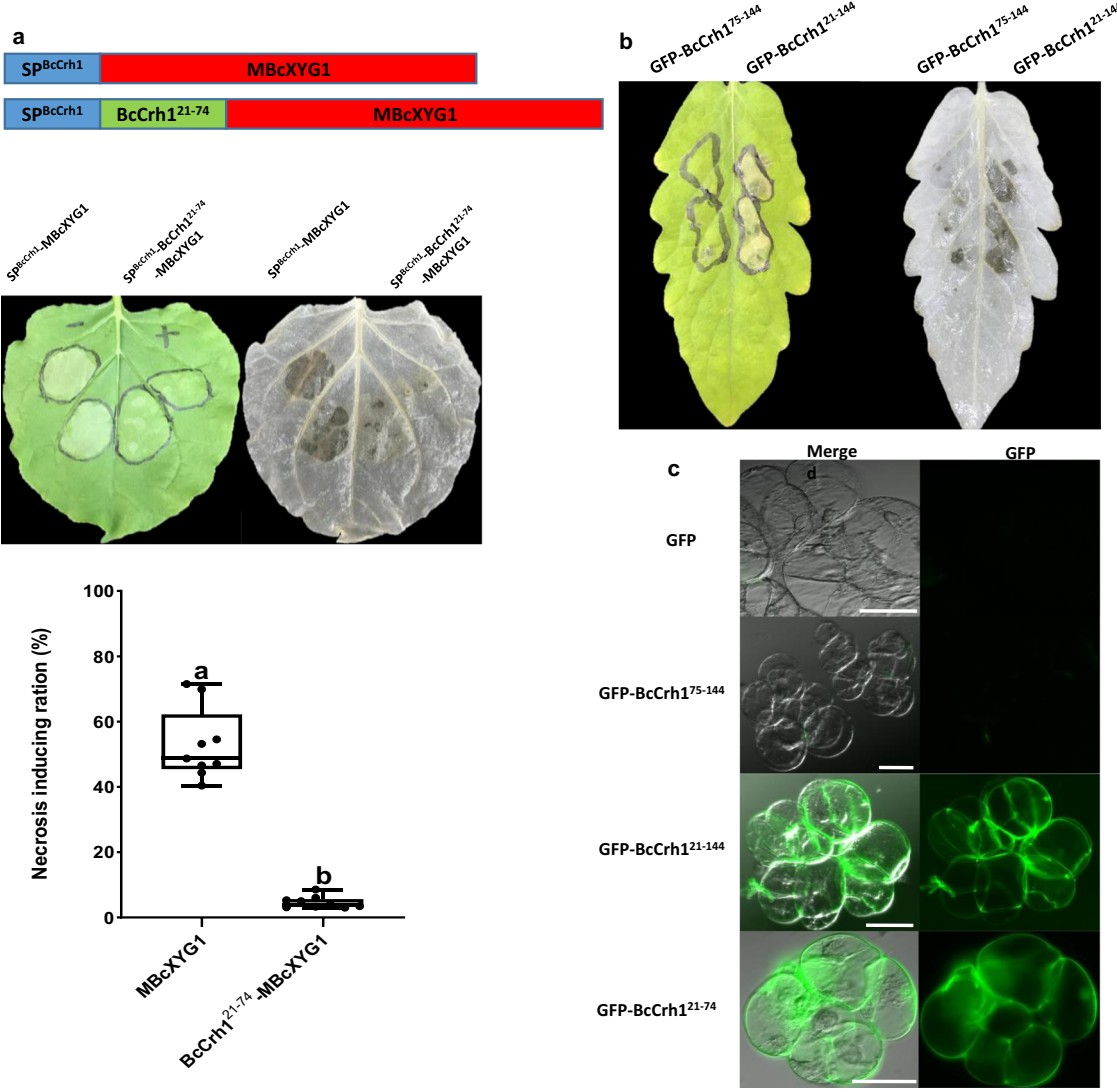

**Fig. 3 A stretch of 53 amino acids at the N′ end of the protein mediates uptake of BcCrh1 by plant cells.** Following analysis of a series of deletion constructs, a minimal region suspected of being necessary for protein uptake was defined. **a** To test the delivery of heterologous proteins using the putative translocation signal, we generated two expression constructs with enzymatic inactive version of the apoplastic NIP BcXYG1 (MBcXyg1) (Zhu et al.[5]). Blue— BcCrh1 SP, Red—MBcXyg1, Green—the suspected delivery sequence. Middle left— images of *N. benthamiana* leaves three days after Agroinfiltration. Leaves were bleached with ethanol (right images) and the necrotic area was calculated. Center lines of the boxplots show the medians, box limits indicate the 25th and 75th percentiles; whiskers cover the full range of values; all data points are plotted as black dots. Data are from 10 sample points from three independent biological replications. Different letters indicate statistical differences at $P \leq 0.001$ ($P = 6.05e-10$) according to unpaired two-tailed Student's *t*-test. **b** Images of tomato leaves 48 h after infiltration with 11 μM of purified BcCrh1-derived peptides, BcCrh1[75–144] and BcCrh1[21–144]. **c** Tomato cell cultures (MsK8) were incubated for 18 h with 5.5 μM solution of GFP-tagged peptides (GFP-BcCrh1[75–144], GFP-BcCrh1[21–144], and GFP-BcCrh1[21–74]), washed three times and visualized by a confocal microscope. Bars = 50 μm. The experiments were repeated three times with similar results.

leaves with purified BcCrh1[21–144] peptide caused significant cell death at 2 dpi, while treatment of leaves with the BcCrh1[75–144] peptide that lacks the putative uptake signal caused no visible symptoms (Fig. 3b). To obtain more direct evidence for the translocation function of the 53 aa region, we generated GFP fusion with the BcCrh1[21–144], BcCrh1[75–144] and BcCrh1[21–74] peptides and administered the purified proteins to a tomato cell suspension-culture. After 18 h of incubation, a GFP signal was detected in the cytoplasmic space of tomato cells incubated with the GFP-BcCrh1[21–144] and GFP-BcCrh1[21–74] protein, but not in cells incubated with the GFP-BcCrh1[75–144] protein (Fig. 3c). Taken together, these results confirmed that BcCrh1 is an intracellular, cell death-inducing effector and that translocation into the plant cells is mediated by the N′ 53 amino acids residues of the protein.

**BcCrh1 triggers PTI and induces host resistance against *B. cinerea*.** Infiltration of *N. benthamiana* with Agrobacterium or purified protein solutions resulted in the accumulation of reactive oxygen species (ROS) and callose deposition (Fig. 4a, b). We also found that tomato defense-related marker genes were activated in response to infiltration with purified protein solutions (Fig. 4c). When leaves were inoculated with *B. cinerea* 48 h after infiltration with the protein, the infection was significantly reduced compared to control leaves pre-treated with GFP protein (Fig. 4d). Thus, along with cell death-inducing activity, BcCrh1 also triggers plant defense responses, similar to the vast majority of *B. cinerea* NIPs[5,12–16,24,25]. Since BcCrh1 does not induce cell death in *A. thaliana* (Supplementary Fig. 4b), we produced *A. thaliana* transgenic lines that express the BcCrh1 protein and tested their sensitivity to Botrytis. The BcCrh1-expressing plants were

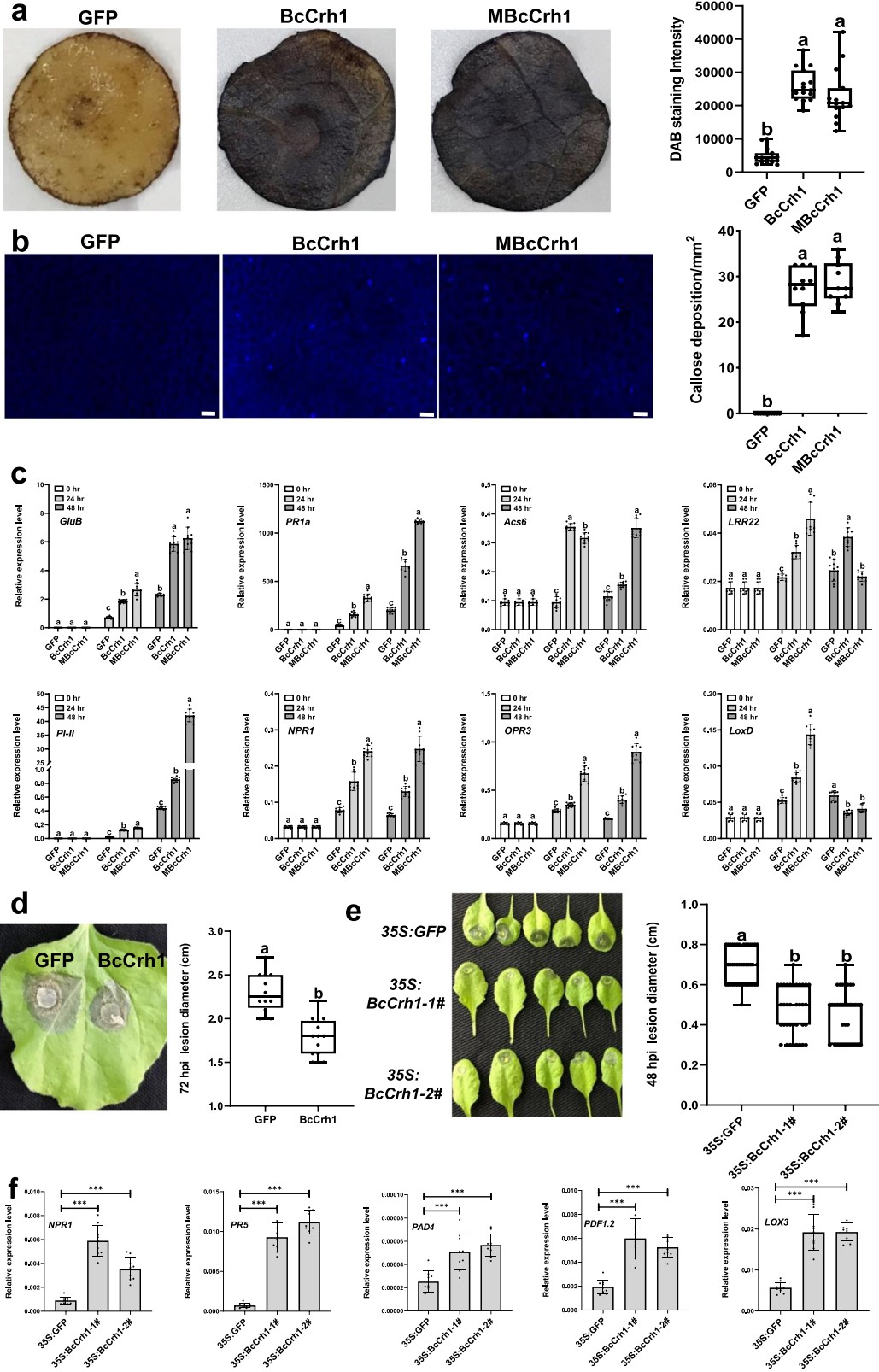

significantly less sensitive to infection by *B. cinerea* than control plants that were transformed with an empty vector, as determined by lesion size 3 dpi (Fig. 4e). Gene expression analysis showed induction of defense marker genes in the BcCrh1-transgenic plants (Fig. 4f), which confirmed that reduced infection was due to induced plants defense.

## BcCrh1 is expressed in planta and secreted from infection cushions.

Crh proteins are fungal cell wall transglycosylases[18–20]. Nevertheless, BcCrh1 was found in the fungal secretome and is internalized by the plant cells, where it triggers plant cell death and defense responses, implying that it also affects fungal-plant interaction. To gain insights into the possible role of BcCrh1 in

**Fig. 4 BcCrh1 triggers plant immunity responses and enhances plant resistance to Botrytis infection. a** ROS accumulation. *N. benthamiana* leaves were Agroinfiltrated with free GFP, and native or enzymatic inactive (MBcCrh1) BcCrh1. After 48 h samples were stained with DAB, they were decolorized with ethanol and staining intensity was quantified with ImageJ. **b** Callose deposition. *N. benthamiana* leaves were infiltrated with 11 μM of purified proteins. Photomicrographs indicating callose deposition induced by BcCrh1 and MBcCrh1. After 24 h leaves were stained with aniline blue, bleached with ethanol, and images were captured with a fluorescent microscope (Bar = 50 μm). Callose levels were estimated by quantification of the number of spots per square millimeter using ImageJ. All data (*n* = 15 in (**a**), *n* = 12 in (**b**)) from three independent biological replicates are plotted as black dots. **c** Defense gene expression. Tomato leaves were infiltrated with purified proteins and relative expression levels of selected defense genes were determined by qRT-PCR 24 after 24 and 48 h. Values represent mean ±;SD (*n* = 9) from three independent biological replicates and three technical replicates. **d** Infection assay. *N. benthamiana* leaves were infiltrated with 11 μM of GFP (mock) or purified proteins, 48 h later the leaves were inoculated with *B. cinerea* mycelia plugs, the plants were incubated for an additional 72 h in a moist chamber and then symptoms were recorded. Data are from three independent experiments, each with four replications. **e** Infection assay of *A. thaliana* transgenic plants. Leaves of transgenic plants that express GFP (*35S:GFP*) or BcCrh1 (two independent plants, *35S:BcCrh1-1#* and *35S:BcCrh1-2#*) were inoculated with droplets of *B. cinerea* spore suspension, the plants were incubated in a moist chamber and symptoms were recorded 48 hpi. Images show all the inoculated leaves from a single plant. Graph shows data of three independent biological replications (*n* = 44, 36, 36). Whiskers of the boxplots in (**a**), (**b**), (**d**), and (**e**) show the minimum and maximum values; center lines of boxplots display the median values; box limits indicate the 25th and 75th percentiles. **f** Defense gene expression of transgenic *A. thaliana* plants. Relative expression levels of defense-related marker genes were measured by qRT-PCR. Values are means ± SD (*n* = 9) from three independent biological replicates and three technical replicates. Asterisks indicate significant difference between control (*35S:GFP*) and BcCrh1 transgenic lines according to one-way ANOVA, *P* ≤ 0.001. In all other graphs, different letters indicate statistical differences at *P* ≤ 0.01 according to one-way ANOVA.

infection development, we studied *bccrh1* gene expression and protein localization in planta. Transcript levels of *bccrh1* increased sharply following infection and reached a peak 12 hpi, in contrast to a much more gradual increase on solid Gamborg's B5 medium that peaked at 60 h post germination (Supplementary Fig. 7). To determine protein localization, a transgenic *B. cinerea* strain was produced with a *bccrh1-gfp* fusion gene under the control of the native *bccrh1* promoter. During saprophytic growth on a culture medium, the protein was observed in the fungal vacuoles and ER, and accumulated to high levels in infection cushions (Fig. 5a, b). Infection assay of onion epidermis cells showed that the protein was secreted from hyphal tip 12–21 hpi, and from infection cushions at 36 hpi (Fig. 5c). Further evidence for BcCrh1 secretion was confirmed by Z-series projection, demonstrating that the GFP-BcCrh1 fusion protein is first delivered to the hyphal tip and later concentrates in infection cushions (Supplementary Fig. 8). Following secretion of the protein from the fungal cells, the GFP signal diffused in the surrounding apoplastic space and then accumulated inside the onion cells, including neighboring cells in regions of the tissue adjacent to the invasion area (Fig. 5d). These results show different expression and localization patterns of BcCrh1 during saprophytic and pathogenic development; During saprophytic growth, the gene is continuously expressed at moderate levels, and the protein is localized inside the fungal cells, while in planta, the gene is highly and transiently expressed following the first contact of the fungus with the plant, and the protein accumulates to high levels in infection structures before being released to the plant apoplast.

**BcCrh1 is dispensable for *B. cinerea* pathogenicity and development**. Deletion or overexpression of the *bccrh1* gene had no visible effect on fungal development or pathogenicity (Fig. 6; Supplementary Fig. 9a–d). To determine whether the enzymatic activity activates a defense response that counteracts the cell death-inducing activity, two additional strains were generated that overexpress the inactive enzymatic protein (MBcCrh1) in wild-type (OE-MBcCrh1) and *bccrh1* deletion (Δ/OE-MBcCrh1) genetic background. Surprisingly, both strains were less virulent than the wild-type strain. In particular, the pathogenicity of the double mutant was severely reduced and symptoms were restricted to local lesions (Fig. 6a, b). Microscopic observation of infected leaves showed the Δ/OE-MBcCrh1 strain had a weaker penetration ability (Fig. 6c, d), which was associated with

impaired infection cushion formation of this mutant compared with other strains (Supplementary Fig. 9d). Furthermore, pathogenicity of the Δ/OE-MBcCrh1 strain was at least partially restored by mechanical injury of leaves prior to infection (Supplementary Fig. 9e), suggesting a specific defect in penetration. The Δ/OE-MBcCrh1 strain also had developmental defects, including reduced sporulation and mycelium production, and an abnormal spore shape (Fig. 6e, f, Fig. 7c, d). However, there was no change in hyphal growth rate and spore germination (Supplementary Fig. 9a, b). These phenotypic changes were all more severe when the *bccrh1* gene was deleted, indicating that they were caused by the accumulation of high levels of the MBcCrh1 protein. To test this hypothesis, we generated a strain that expresses the MBcCrh1 protein under control of the native *bccrh1* promoter in the background of Δ*bccrh1* (strain Δ/NP-MBcCrh1). This strain did not show any developmental defects and it formed normal infection cushions and disease symptoms (Fig. 7), confirming that the developmental defects of the Δ/OE-MBcCrh1 strain resulted from high level accumulation of the enzyme inactive protein. The results also showed that the pathogenicity defects were caused at least in part by defects in plant invasion due to deformed infection cushions. Based on these findings, we predicted that BcCrh1 forms protein dimers necessary for the transglycosylase activity.

**BcCrh1 forms homodimers as well as heterodimers with other BcCrh protein members**. A yeast two-hybrid assay showed that both the native and the inactive BcCrh1 enzyme form protein dimers (Fig. 8a). The dimerization of BcCrh1 was further confirmed in vitro by a pull-down assay, in planta by Co-immunoprecipitation (Co-IP) and in vivo by BiFC (Fig. 8b–d). Sequence alignment of fungal Crh proteins revealed two conserved cysteine residues at positions 26 and 33 (Supplementary Fig. 1c), which form intramolecular disulfide bonds that might be important for proper folding and stability of the protein (Supplementary Fig. 2). Site-directed mutagenesis of either or both $C^{26}$ and $C^{33}$, as well as deletion of the sequence including $C^{26–33}$, all showed that $C^{33}$, but not $C^{26}$, was required for BcCrh1 homo-dimer formation (Fig. 8e). Despite the importance of $C^{33}$ for homodimer formation and protein structure, the BcCrh1$^{C26AC33A}$ mutant protein retained full necrosis-inducing activity (Supplementary Fig. 3b, Supplementary Fig. 5), implying that the monomeric peptide was sufficient for induction of cell death, independent of the protein tertiary structure. However, the BcCrh1$^{C26AC33A}$ did not

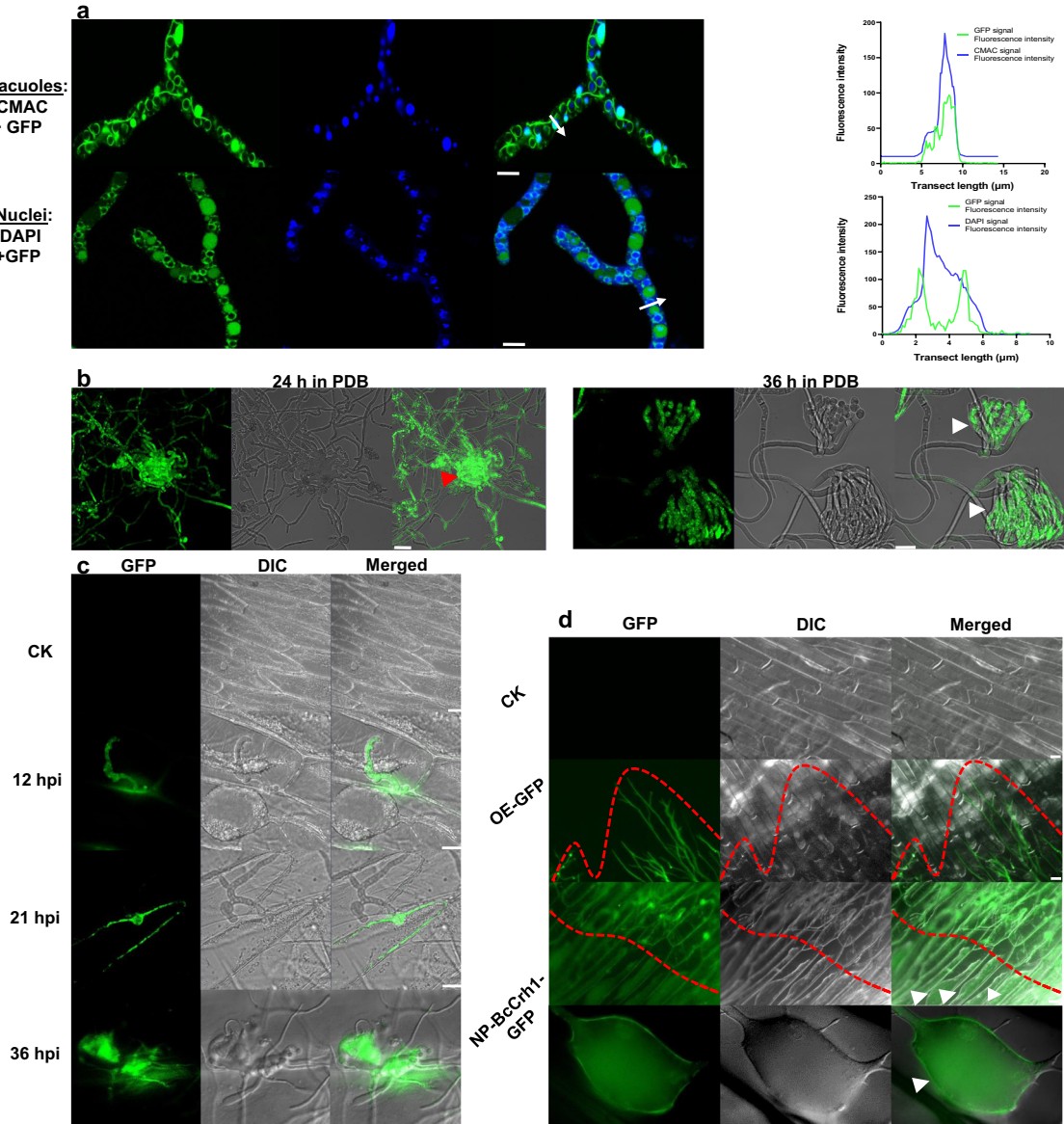

**Fig. 5 Subcellular localization of BcCrh1 during saprophytic growth and host infection. a** Intracellular localization of BcCrh1 during saprophytic growth. Spores of BcCrh1-GFP strain were cultured in liquid PDB medium for 12 h, vacuoles (top) and nuclei (bottom) were stained with CMAC and DAPI, respectively, and samples were visualized using a Confocal microscope. Scale bars = 5 μm. The graph shows fluorescent intensity profiles of GFP/CMAC signals (top) and GFP/DAPI signals (bottom) in transects (white arrowheads). Y axis, GFP and CMAC or DAPI fluorescence intensity; X axis, transect length (μm). **b** Differential distribution of BcCrh1 in mycelia and infection cushions in vitro. Spores were germinated on a glass slide in PDB medium. At 24 h the GFP signal accumulates in the entire mycelium and in initiating infection cushions (left, marked with red arrow), at 36 h the signal is detected only in the mature infection cushions (right, indicated with white arrows). Scale bar = 20 μm. **c–d** Onion epidermis infection assay with cytoplasmic GFP and BcCrh1-GFP strains. **c** Images showing secretion of the protein from hyphal tips at early time points (12, 21 hpi) and from infection cushions at 36 hpi. Scale bars = 50 μm at 12 hpi and 20 μm in all other images. **d** Intracellular localization of secreted BcCrh1-GFP protein in plasmolysed onion cells 45 hpi. Infection area is marked by red dashed line. BcCrh1-derived GFP signal is observed in the cytoplasmic space of plasmolysed onion cells outside the invasion area (marked with arrows). Scale bars = 20 μm. All the experiments were repeated three times with similar results.

complement the Congo red hypersensitivity of the yeast *crh1Δcrh2Δ* double mutant, supporting the notion that protein dimerization may be necessary for the transglycosylation activity (Supplementary Fig. 6). Yeast two-hybrid assays with the three other BcCrh members showed that BcCrh2 and BcCrh3, but not BcCrh4, can also form homodimers, and that BcCrh1, BcCrh2 and BcCrh3 can form heterodimers with each other, all of which were independent of the enzymatic activity and mediated by the conserved cysteine residues (Fig. 8e, f). To verify our hypothesis that the excessive MBcCrh1 disrupts the function of the native *B. cinerea* Crh proteins by formation of enzymatic inactive protein

dimers, we overexpressed the MBcCrh1$^{C26AC33A}$, which is unable to form dimers (Fig. 8d, e), in a Δ*bccrh1* background. In accordance with our prediction, this strain had no growth defects, and it caused normal disease symptoms (Fig. 7).

## Discussion

GH are the largest GO group in *B. cinerea* secretome. The majority of the GHs are hydrolases of plant cell wall sugar polymers, such as cellulose, pectin, hemicellulose and other plant cell wall-associated polysaccharides[26]. Among them, a small number of proteins (catalytic NIPs) can also trigger cell death in

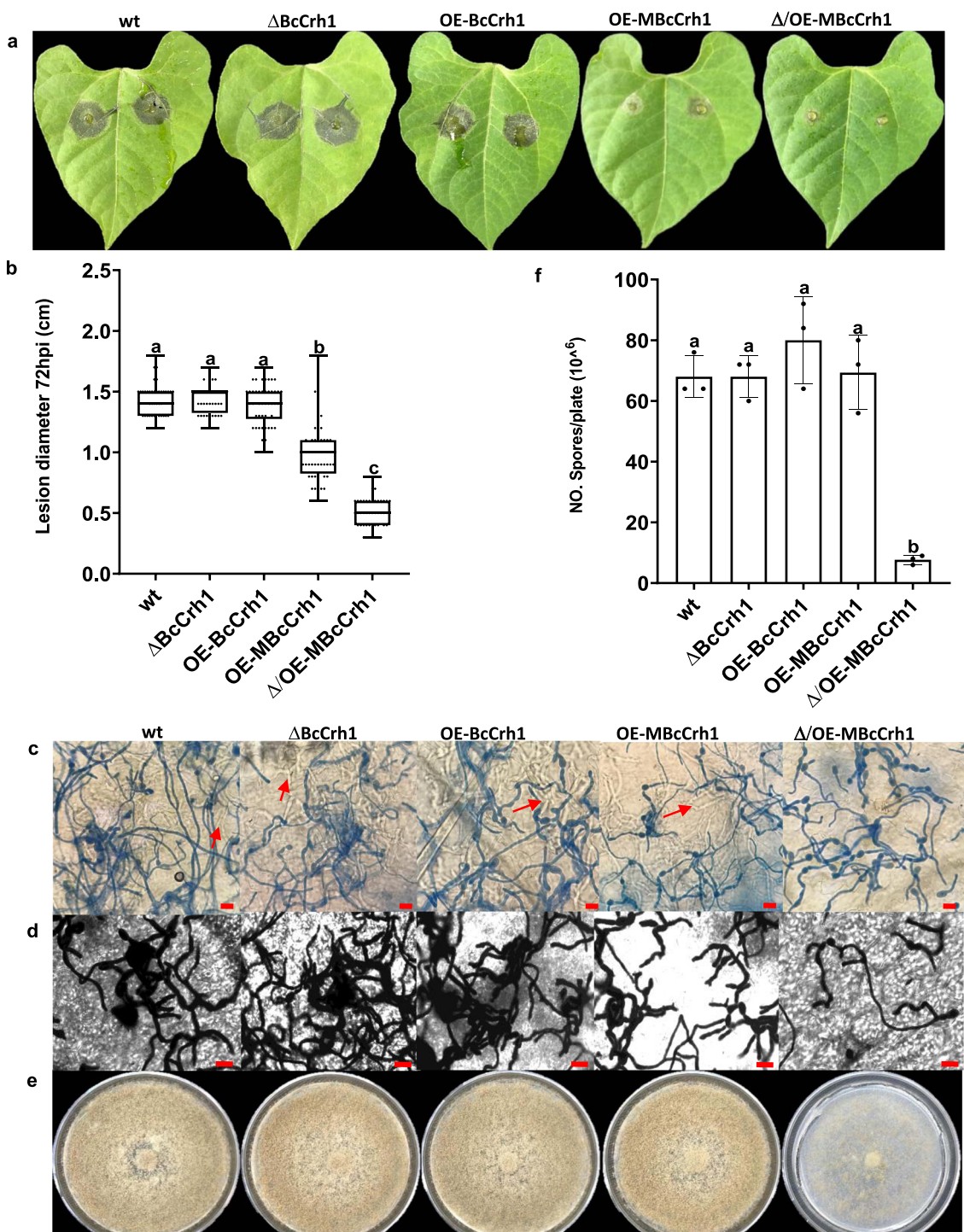

**Fig. 6 Over expression of MBcCrh1 causes reduced pathogenicity and developmental defects. a–b** Infection assay. Bean leaves were inoculated with spore suspensions of *B. cinerea* wild type (wt), *bccrh1* deletion (ΔBcCrh1), *bccrh1* overexpression (OE-BcCrh1), and strains over-expressing the enzyme inactive (MBcCrh1) protein in wild type (OE-MBcCrh1) or *bccrh1* deletion (Δ/OE-MBcCrh1) genetic background. Symptoms were photographed and the lesion diameter was recorded 72 hpi. Box limits show the 25th and 75th percentiles. The center lines of boxplots indicate the medians values; whiskers extend to minimum and maximum values from the 25th and 75th percentiles; all data are indicated as black dots. At least 40 sample points from three independent biological replicates were used for statistical analysis. **c–d** Lactophenol cotton blue and lactophenol trypan blue staining of infected leaves. Leaf tissue was harvested at the designated time points and stained with lactophenol cotton blue, which stains only the exposed hyphae (**c**), and with lactophenol trypan blue, which stains both exposed and intracellular (red arrows) hyphae (**d**). Bar = 20 μm. Note the near absence of intracellular hyphae in the Δ/OE-MBcCrh1 mutant, which indicates penetration defects. All the experiments were repeated three times with similar results. **e–f** Mycelium and spore production. Fungi were cultured on solid GB5-Glucose medium and grown at 20 °C with continuous light. Pictures were taken (**e**) and spores counted after eight days of incubation. Data represent mean ± SD from three independent biological replicates. Different letters in (**b**) and (**f**) indicate statistical differences at $P \leq 0.01$ according to one-way ANOVA.

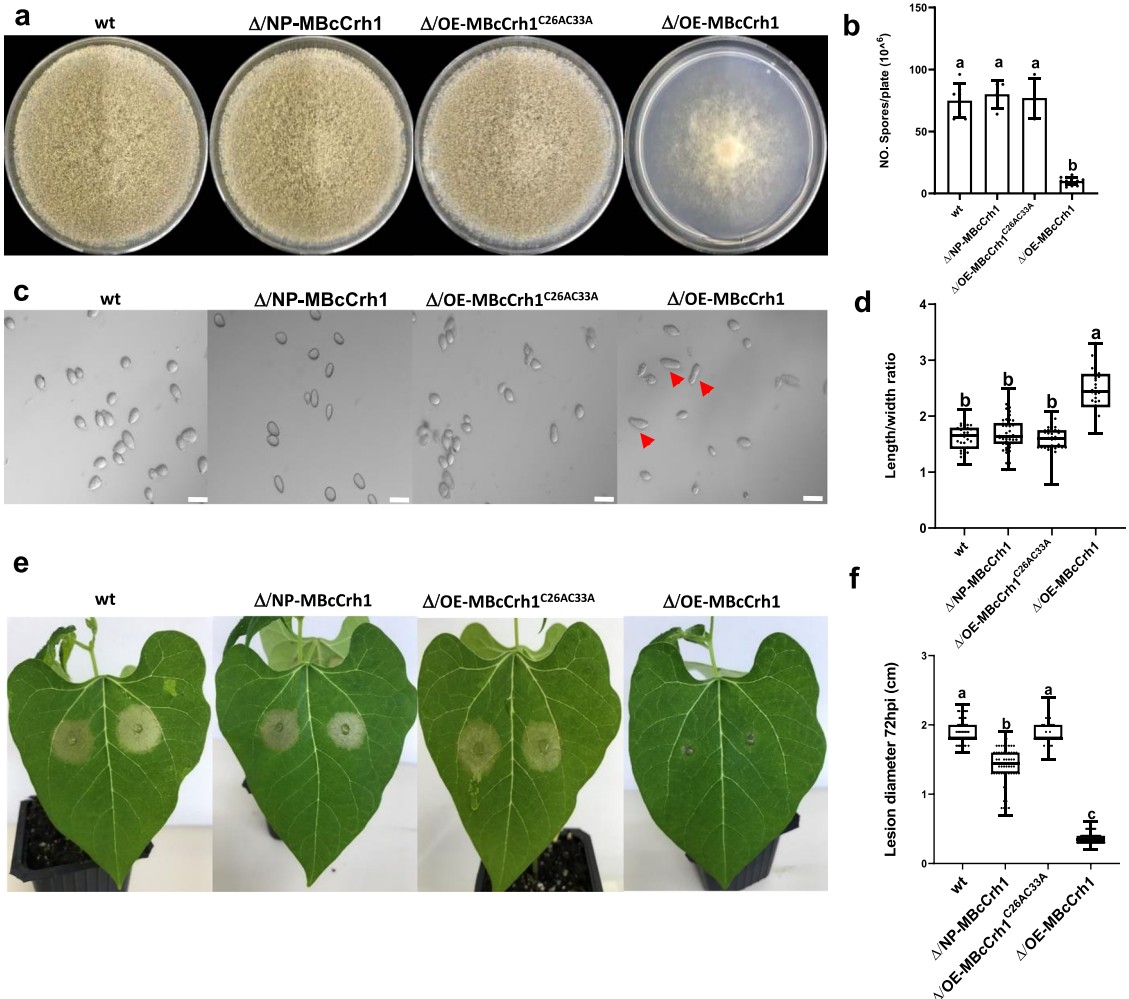

**Fig. 7 The pathogenicity and developmental defects induced by MBcCrh1 depend on expression levels.** The following strains were characterized: *B. cinerea* wild type (wt), enzyme inactive BcCrh1 with the native promoter in the background of *bccrh1* deletion (Δ/NP-MBcCrh1), overexpression of MBcCrh1 with mutation of C26 and C33 in a background of *bccrh1* deletion (Δ/OE-MBcCrh1^C26AC33A), and overexpression of MBcCrh1 in a background of *bccrh1* deletion (Δ/OE-MBcCrh1). **a**–**d** Spore production and shape. Fungi were cultured on solid GB5-Glucose medium and grown at 20°C with continuous light. Pictures of plates (**a**) and spores (**c**) were taken after eight days of incubation. Spores were harvested and average spore numbers (**b**) and spore dimensions (**d**) were determined. Arrows indicate spores of the Δ/OE-MBcCrh1 strain with abnormal morphology. For spore numbers, data represent mean ± SD from three independent biological replications. For spore dimensions, the ratio of length/width was calculated. Data of at least 30 spores from three independent biological replications were used for statistical analysis. **e-f** Bean leaves were inoculated with spore suspensions of the different strains, pictures were taken and lesion size recorded 72 hpi. At least 32 sample data from three independent biological replications were used for statistical analysis. In boxplots (**d** and **f**), center lines represent the medians, box edges show the 25th and 75th percentiles; whiskers extend to minimum and maximum values from the 25th and 75th percentiles; all present data are indicated as black dots. Different letters in d and f indicate statistical differences at $P \leq 0.01$ according to one-way ANOVA.

plants[5,12–14], similar to the non-catalytic NIPs, such as NEP/NELP[6,7] and ceratoplatanins[25,27]. Catalytic NIPs that have been characterized so far remain in the apoplast after secretion by the fungus, and their cell death-inducing activity is mediated by plant extracellular membrane components[5,12,13]. The BcCrh1 protein reported here represents a new class of NIPs: it has a defined catalytic domain but lacks a plant cell wall degrading activity, and its site of action is inside, rather than outside the plant cell. Furthermore, the cell death-inducing activity of BcCrh1 is unexpected and surprising, since Crh family proteins catalyze crosslinking of sugar polymers in the fungal cell wall. Thus it lacks a clear connection to host-fungal interaction.

The Crh protein family is a conserved group of fungal-unique enzymes catalyzing cell wall maturation steps like chitin-glucan and chitin-chitin crosslinking[18–20]. Deletion of *S. cerevisiae CRH1* and *CRH2* genes affected cell wall integrity and proper

development, particularly upon cell wall stress. Similar results were obtained in other yeast species[18,19,28,29], while deleting the five *crh* gene members in *A. fumigatus* had almost no effect on fungal development[20]. Similarly, deletion of *bccrh1* did not affect *B. cinerea* development and only slightly increased sensitivity of the fungus to Congo red (Supplementary Fig. 9c). However, we found that the BcCrh proteins form dimers, and this dimerization seems to be necessary for the transglycosylase activity. Over-production of the enzyme inactive monomer resulted in severe developmental defects, indicating that two intact monomers are necessary for proper function, and further supporting a redundant role of Crh proteins in filamentous species. These results reveal a new aspect in Crh proteins function, potentially leading to a better understanding of their mechanistic mode of action.

Under saprophytic conditions, the BcCrh1 protein was localized inside vacuoles and the ER. Still, during pathogenic

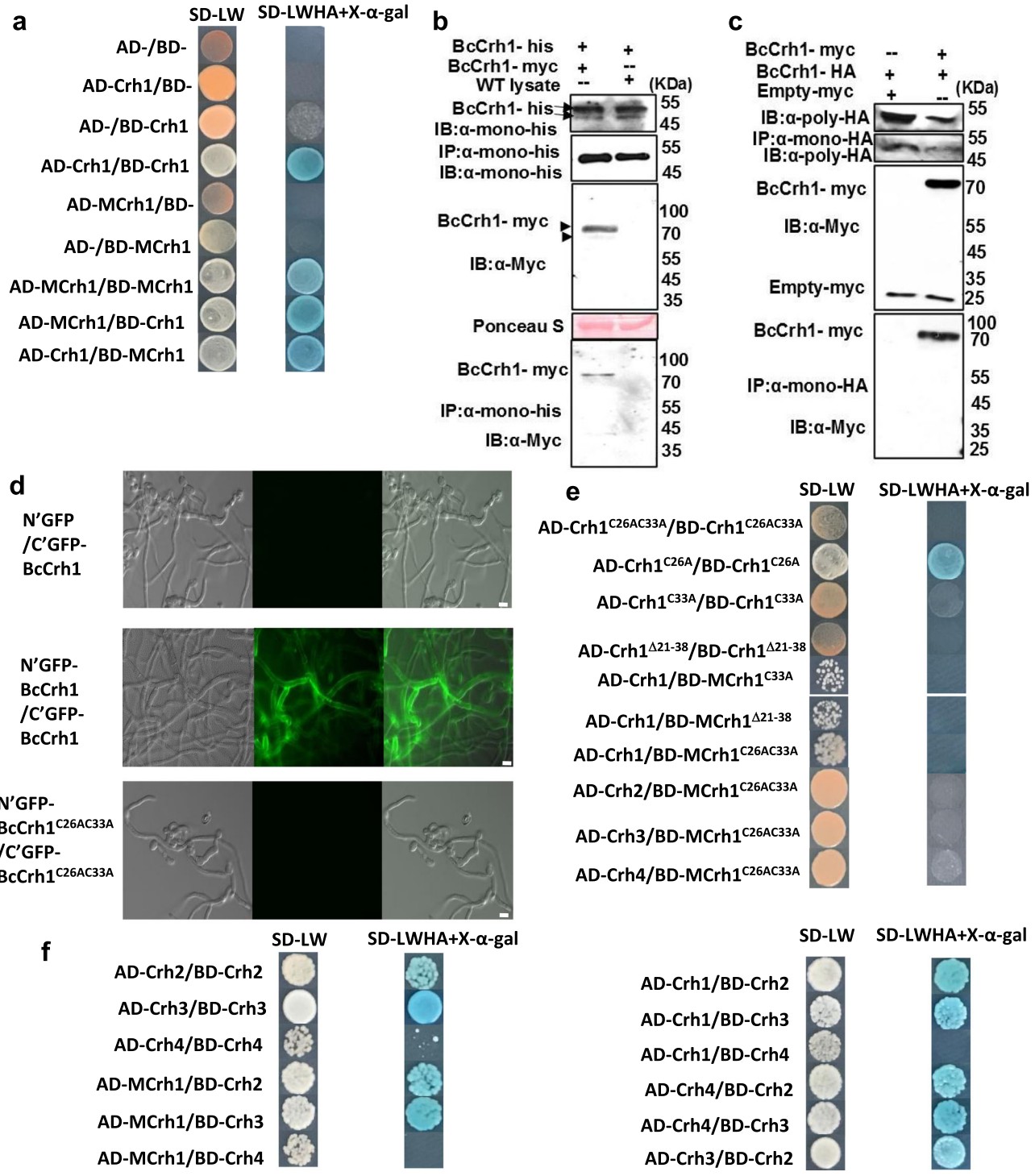

**Fig. 8 BcCrh protein dimerization. a** Yeast two-hybrid assay (Y2H) of BcCrh1 and MBcCrh1 homodimer formation. SD/-Trp-Leu medium was used to confirm the transformation events. SD/-Trp-Leu-His-Ade medium containing X-α-gal was used to screen yeast strains with the positive protein-protein interaction. **b–d** Confirmation of BcCrh1 dimer formation by in vitro pull down assay (**b**), Co-immunoprecipitation (Co-IP) assay (**c**) and bimolecular fluorescence complementation (BiFC) assay (**d**). **b** For the in vitro pull-down assay, BcCrh1-myc was transiently expressed in *N. benthamiana* and pull down was performed with BcCrh1-His recombinant protein conjugated with histidine beads. Protein from non-infiltrated leaves (WT lysate) was used as negative control. **c** For the Co-IP experiment, BcCrh1-myc or empty-myc were co-expressed with the BcCrh1-HA in *N. benthamiana*. BcCrh1-myc, but not empty-myc, co-immuno-precipitated with BcCrh1-HA conjugated to HA beads. **c** BiFC was conducted with *B. cinerea* transgenic strain expressing split GFP-BcCrh1 proteins (N'GFP-BcCrh1/C'GFP-BcCrh1). A strain expressing split GFP fused to BcCrh1$^{C26AC33A}$ (N'GFP-BcCrh1$^{C26AC33A}$/C'GFP-BcCrh1$^{C26AC33A}$) did not show any fluorescence, indicating a lack of dimer formation by this protein derivative. Bar = 10 μm. The experiments were repeated three times with similar results. **e** Effect of point mutations or deletion at the N' part of BcCrh1 on homo- and heterodimer formation. **f** Y2H assay of interaction between different BcCrh members.

development, high levels of the protein were observed in developing infection cushions, before being secreted to the plant apoplast. The fungal strains in which BcCrh dimerization was blocked exhibited defective infection cushions and drastically reduced pathogenicity. This specific accumulation of BcCrh1 in infection cushions and aberrant infection cushions in the mutant suggests a possible role for BcCrh1 in infection cushion formation, which is associated with retardation of hyphal extension and hyper branching. It is possible that, following these pathogenicity-specific developmental events, the excessive enzyme, now serving as a virulent factor, is released from the mature infection cushions and induces plant cell death.

BcCrh1 is a cytoplasmic effector, unlike previously isolated catalytic NIPs, which remain in the plant apoplast. Delivery of the protein to the plant nucleus (by addition of NLS) prevented cell death, indicating that the site of action is in the plant cytosol (Supplementary Fig. 3c). The uptake of BcCrh1 by plant cells is pathogen-independent and mediated by the first 53 aa of the mature BcCrh1 peptide. This feature places BcCrh1 in the category of cell-penetrating peptides (CPPs), a class of short peptides that facilitate cellular uptake of various molecules through endocytosis, macropinocytosis, and direct plasma membrane penetration[30]. Compared with animals, only a small number of CPPs have been reported in plants, including several that mediate effectors uptake, mainly from oomycetes. Specific motifs that have been associated with the delivery of oomycetes effectors include the RXLR motif common in *Phytophthora* effectors[31–34], LXLFLAK in CRN proteins, and the CHCX-containing amino terminus motif in *Albugo laibachii* effector proteins[35–37]. RXLR effectors are assumed to be internalized via endocytosis in a pathogen-independent manner[38,39], whereas the mechanisms of uptake of CRN and CHXC effectors remain elusive[33]. A relaxed RXLR-like motif has been proposed to be involved in the pathogen-independent, PI3P-mediated endocytosis of certain fungal effectors[39–41]. However to date, there is no known conserved motif that is shared by candidate cytoplasmic fungal effectors[33,42]. Similar to other fungal cytoplasmic effectors, the BcCrh1 translocation sequence does not contain any known conserved motifs and it lacks homology to sequences that mediate translocation of effectors in other fungi. Thus, the N-terminal uptake signal of BcCrh1 may be considered another class of CPP.

Similar to all other NIPs, along with plant cell death, BcCrh1 also elicits a PTI response. The cell death and PTI have opposite effects on disease development, which together with high redundancy of NIPs precludes a clear phenotype in deletion or over-expression fungal strains. However, BcCrh1 did not induce cell death in *Arabidopsis*, which enabled a better assessment of the induced PTI on disease development. Indeed, transgenic *Arabidopsis* plants that produce BcCrh1 were considerably less sensitive to *B. cinerea*, demonstrating the potential of BcCrh1, and possibly other NIPs, in engineering pathogen resistance in plants.

## Methods

**Fungi, bacteria, yeasts, and plants**. *Botrytis cinerea* B05.10 and derived transgenic strains were routinely cultured on potato dextrose agar medium (PDA, Acumedia) at 22 °C under continuous fluorescent light supplemented with near UV (black) light. The transgenic fungal strains were cultured on PDA amended with 100 μg/ml hygromycin B (Calbiochem) and/or 100 μg/ml Nourseothricin (Sigma-Aldrich). *Escherichia coli* strain DH5α and BL21 (DE3) were used for plasmid construction and propagation, and for protein expression, respectively. *A. tumefaciens* strain GV3101 was used for *A. tumefaciens*-mediated transient expression of proteins in plant leaves. All the bacteria were grown at 37 °C on LB agar plates or in LB liquid medium supplemented with 100 μg/ml ampicillin and 50 μg/ml kanamycin. Yeast strain AH109 was used for yeast two-hybrid assay. For yeast complementation assays, *S. cerevisiae* BY4741 and GRA007[43] strains were used. French bean (*Phaseolus vulgaris* L. genotype N9059), *N. benthamiana*, *A. thaliana* (ecotype Columbia-0), tomato (*Solanum lycopersicum*) cv. Hawaii

7998, and maize (*Zea mays*) cv. silver queen were grown in a growth room at 20 °C (*A. thaliana*), or 25 °C (all other plant species) with 16-h/8-h light/dark cycle.

**Bioinformatics analysis**. The genomic sequence database of *B. cinerea* (https://mycocosm.jgi.doe.gov/Botci1/Botci1.home.html) was used for Blast searches of *B. cinerea* genes. The SignalP 5.0 server (http://www.cbs.dtu.dk/services/SignalP/) was used to predict the presence of signal peptides and the location of their cleavage sites in the proteins[44]. TMHMM Server v. 2.0 (http://www.cbs.dtu.dk/services/TMHMM/) was used for the prediction of transmembrane helices in proteins[45]. The conserved protein domain search was performed by SMART MODE (http://smart.emblheidelberg.de/smart/change_mode.pl)[46], while the databases NCBI and UniProt were used for BLASTp searches. Multiple Sequence Alignment (MSA) was performed by Clustal Omega (https://www.ebi.ac.uk/Tools/msa/)[47] and MView Version 1.63 was used to present the result. HHpred (https://toolkit.tuebingen.mpg.de/tools/hhpred)[48] was used for the prediction of the 3D structural models of the template from PDB database. The template with the highest sequence identity was used for modeling. To produce the pairwise alignment between the two proteins, ConSurf was used to collect homologues[49]. Sequences were collected from the clean UniProt database with 95% maximal identity between sequences and minimal 35% identity for homologues. Sequences that introduced large gaps into the alignment were discarded, and the pairwise alignment of BcCrh1 and 6IBW was deduced from the multiple sequence alignment. MODELLER (https://salilab.org/modeller/) was used to produce different models[50], and each model underwent a short energy minimization using GROMACS and the AMBER99SB-ILDN force field (https://www.nvidia.com/es-la/data-center/gpu-accelerated-applications/gromacs/)[51,52]. Finally, the model result with the predicted lowest energy was chosen. The models cover amino acids 25 to 275 (according to the amino acids numbering in UniProt entry A0A384J6C4). Each model underwent a short energy minimization using GROMACS and the AMBER99SB-ILDN force field. To test potential dimerization interfaces, PISA (https://www.ebi.ac.uk/pdbe/pisa/)[53] and the template's crystal structure (6IBW) were used. PyMOL 2.4 (https://pymol.org/2/) for visualization of 3D structural models.

**Construction of plasmid DNA**. Primers used in this study are listed in Supplementary Table S1. All constructs were sequence-verified to confirm their integrity before further manipulation. Binary plasmids based on 2 × 35S-MCS-eGFP (pCNG) were used for gene expression in plants (*N. benthamiana* and *A. thaliana*)[54]. The ORFs of target genes were amplified from cDNA with Phusion High-Fidelity DNA Polymerases (NEB). Signal peptide sequence deletion, C-terminus truncations, the addition of nuclear localization (NLS) or myristoylation (CBL) signals, and site-directed PCR mutagenesis were conducted on the target sequences according to specific requirements. The fragments were cloned into linearized pCNG (the vector plasmid was excised with *XbaI* and *BamHI*) and fused to the eGFP using an *E. coli* DH5α-mediated DNA assembly method[55]. For the generation of *B. cinerea* mutant strains, several plasmids were constructed. For deletion of the *bccrh1* gene, 500 bp each of *bccrh1*-5′ and *bccrh1*-3′ flanking regions were amplified from B05.10 genomic DNA and added on each side of a hygromycin-resistance cassette to produce the *bccrh1* deletion plasmid pTZ-Δ*bccrh1*. To construct the *bccrh1* over-expression vector, the full-length *bccrh1* ORF fused to an HA tag at the C-terminus was cloned into the pH2G vector[5]. To generate an enzyme inactive BcCrh1 protein, the catalytic residues were mutated (E120Q/D122H/E124Q) by site-directed PCR mutagenesis using template cDNA, and the amplification fragment was cloned into pJET plasmid to generate the pJET-MBcCrh1 vector. For generation of fungal strains that overexpress the MBcCrh1 in a wild type and *bccrh1* deletion background, the M*bccrh1* clone was introduced into pNAN-OGG (contains a nourseothricin resistance cassette)[56], between the *Aspergillus nidulans* POliC promoter (for overexpression) or the *bccrh1* promoter, and *B. cinerea* Tgluc terminator. Mutagenesis of the conserved cysteine residues at the N′ end (C26A/C33A/E120Q/D122H/E124Q) were generated by site-directed PCR mutagenesis and the amplified fragments were cloned into a pNAN-OGG vector. For the BiFC assay in *B. cinerea*, split-GFP construction (N′GFP-BcCrh1/C′GFP-BcCrh1 under the control of bidirectional H2B promoter) was generated. For protein expression and purification, the sequence encoding mature proteins without the signal peptide was cloned into pET-14b (+) (Novagen) to generate the expression vector pET14b-6xHis-BcCrh1. Expression vectors of truncated-BcCrh1 were constructed in a similar way. To generate the constructs used for a yeast-two-hybrid assay (Y2H), the ORFs of target genes without the secretion sequence were amplified using PCR or site-directed mutagenesis PCR (C26A, C33A, C26A/C33A, and Δ21–38), and the PCR amplification products were cloned into the pGADT7 and pGBKT7 yeast two-hybrid vectors. To generate the vectors used for yeast complementation assay, the ORFs of target genes were introduced into YEp352 under the control of *CRH2* promoter and terminator[43].

**DNA and RNA analyses**. Genomic DNA and total RNA extractions were performed by Extract-N-Amp™ Tissue PCR Kits (Sigma/aldrich) and TRIzol reagent (Sigma/Invitrogen). For cDNA synthesis, total RNA was treated with DNase I (Thermo Scientific) and then the first-strand cDNA was synthesized from 1 μg of DNA-free RNA using the RevertAid First Strand cDNA Synthesis Kit (Thermo Scientific). qRT-PCR was performed with SYBR Premix Ex Taq II (Takara, Dalian,

China) using Mx3000P (Stratagene, La Jolla, CA, USA) or StepOne (Applied Biosystems) Real-time PCR instruments. Relative fold change of mRNA levels was normalized with the *B. cinerea* *bcgpdh* (BC1G_05277), *S. lycopersicum* actin gene, or *A. thaliana* UBQ10 gene (AT4g05320).

**A. tumefaciens-mediated transient expression**. Constructs were introduced into *A. tumefaciens* strain GV3101 by electroporation, and Agrobacterium infiltration assays were performed on *N. benthamiana* plants that were grown for 4–5 weeks in the greenhouse as described previously[57]. Briefly, *A. tumefaciens* strains were inoculated into 1 ml of LB liquid medium containing 50 μg/ml of kanamycin and 50 μg/ml of rifampin. The bateria were incubated at 28 °C overnight, the cells were harvested by centrifugation at 2000 × g for 5 min and then resuspended in the infiltration buffer (10 mm MgCl2, 10 mm MES, pH 5.6, 100 μM acetosyringone) and diluted to OD600 = 0.5–1. Unless otherwise mentioned, leaves were photographed five days after infiltration. Expression of all the proteins was verified by western blot analysis 2–3 d after Agrobacterium infiltration (Supplementary Fig. 11).

**Leaf infiltration assay with purified proteins**. To test the cell death-inducing activity of recombinant proteins, 2.75–55 μM protein solutions were infiltrated into leaves of *N. benthamiana*, and leaves of *S. lycopersicum*. Leaves of *A. thaliana* and *Z. mays* were infiltrated with 11 μM protein solution. Infiltrated leaves were photographed 2–3 days after infiltration. To test the induced plant defense responses, total RNA was extracted from *S. lycopersicum* leaves at 0, 24, and 48 h after infiltration with recombinant protein solutions (11 μM) and qRT-PCR analysis was performed for measurement of plant defense-related gene expression levels. The leaves of *N. benthamiana* were infiltrated with 11 μM purified proteins, then sampled at 24 h for quantification of callose deposition.

**Uptake assay of GFP-labeled proteins by plant cells**. Tomato cell cultures (MsK8) were incubated for 30 min with incubation buffer (10 mM phosphate buffer, pH 7, and 0.1% BSA), followed by the addition of GFP-tagged peptide solutions (5.5 μM) for 18 h. The cells were subsequently washed three times in incubation buffer and visualized with confocal laser microscopy.

**Protein analysis**. Plasmids used for protein expression were transformed into *E. coli* strain BL21 (DE3). Expression and purification of the recombinant proteins were performed using Ni-NTA resin (GE Healthcare) according to previously described protocol[58]. In brief, 500 mL of LB medium (supplemented with 100 μg/ml ampicillin) was inoculated with 5 ml of pre-cultured *E. coli* BL21 strain containing the desired plasmid and the cultures were incubated at 37 °C with shaking until they reached OD$_{600}$ = 0.8. The culture was cooled to 16 °C, IPTG (isopropyl-β-D-thiogalactoside) was added to a final concentration of 0.2 mM, and the cultures were incubated with shaking overnight. Cells were harvested by centrifugation at 5,000 g for 10 min and the pellet was resuspended in buffer A (20 mM sodium phosphate, 300 mM sodium chloride (PBS) with 10 mM imidazole; pH 7.4). The cells were busted by sonication, centrifuged at 23,000 × g for 30 min at 4 °C, and the supernatant was loaded onto a Ni-NTA column. The columns was washed with buffer B (PBS with 25 mM imidazole; pH 7.4), and eluted with buffer C (PBS with 250 mM imidazole; pH 7.4). The protein was further cleaned and concentrated by using a 10-kDa molecular weight cut-off Amicon Ultra centrifugal filter (Millipore).

For analysis of proteins following Agroinfiltration, leaves were grounded in extraction buffer [20 mM Tris-HCl, pH 7.5, 100 mM NaCl, 1 mM EDTA, 2 mM NaF, 1 mM Na₃VO₃, 1 mM dithiothreitol (DTT), 0.5% Triton X-100, 10% glycerol and 1 × protease inhibitor cocktail] and the extract was centrifuged at 12,500 × g for 20 min at 4 °C. The cell lysate was harvested, boiled with 4 × SDS protein loading buffer at 95 °C for 15 min, and detected by immunoblotting with indicated antibodies.

**Fluorescence and confocal microscopy**. Samples were collected from *N. benthamiana* leaves two days after Agroinfiltration. For plasmolysis, samples were infiltrated with 0.8 M mannitol solution for 20 min. For subcellular localization of GFP-labeled BcCrh1 in fungal cells during saprophytic growth, conidia were suspended in PDB and 10 μl droplets of spore suspension containing 1 × 10⁴ conidia ml⁻¹ were placed on a glass slide and incubated at 20 °C. Confocal microscopy was performed with a Zeiss LSM780 confocal microscope system and ZEISS ZEN 3.0 (blue edition) imaging Software[59]. Epifluorescence and light microscopy were performed with a Zeiss Axio imager M1 microscope and Carl Zeiss AxioVision Rel. 4.8 Software. Differential interference microscopy (DIC) was used for bright field images. DAPI filter (340–390 and 420–470 nm excitation and emission, respectively) was used for visualization of DAPI-stained nuclei and CMAC stained vacuoles. eGFP and mCherry fluorescence were collected using excitation laser wavelengths of 488 and 561 nm, respectively. For GFP, emission was collected in the range of 493–535 nm. Images were captured with a Zeiss AxioCam MRm camera.

**Transformation of *B. cinerea* and characterization of the transformants**. PEG-mediated protoplast transformation of *B. cinerea* was performed according to published protocols[60]. Transgenic strains are listed in Supplementary Table S2. At least three independent single spore isolates from independent colonies were obtained for each strain. The ΔBcCrh1 colonies were confirmed by PCR to verify deletion of the *bccrh1* gene and the level of *bccrh1* gene in expression in over-expression strains was determined by qRT-PCR. The phenotypic assays, including mycelial growth rates, conidiation, conidial germination and stress tolerance, were performed according to a previous description[60,61].

**Pathogenicity and infection-related assays**. The pathogenicity assay with *B. cinerea* was performed as described previously[60]. For bean plants, the first two primary leaves of a 10-day-old plants were inoculated with 7.5 μl of conidia suspension containing 2 × 10⁵ conidia ml⁻¹. For *A. thaliana* plants, leaves of 4–5-week-old plants were similarly inoculated with 5 μl conidial suspension. Infection intensity was evaluated by measurement of lesion diameter 72 hpi. For the infection cushion formation assay, conidia were suspended in GB5 + 2% Glucose, 10 μl were placed on glass slides and incubated in a moistened chamber at 20 °C for 36 h. For localization of GFP-labeled BcCrh1 during infection, conidia were suspended in PDB medium and 10 μl droplets of spore suspension containing 1 × 10⁴ conidia ml⁻¹ were placed on onion epidermal cells and incubated at 20 °C in moist conditions. Samples were prepared for microscopy observation at designated time points.

**Staining dyes and procedures**. Cotton blue and trypan blue staining of fungal hyphae for the penetration assay of plant leaves were performed according to published procedures[62]. DAB staining of ROS and aniline blue staining of callose depositions were conducted based on the published descriptions[63,64]. Briefly, detached plant leaves were soaked overnight in one of the following staining solutions: lactophenol cotton blue (0.5 mg/ml of cotton blue dissolved in lactophenol containing an equal volume of lactic acid, glycerol, liquid phenol and ddH₂O), trypan blue (2.5 mg/ml of trypan blue dissolved in lactophenol containing an equal volume of lactic acid, glycerol, liquid phenol and ddH₂O), DAB (1 mg/ml of DAB dissolved in ddH2O, pH 3.6), or aniline blue (10 mg/ml of aniline blue in 150 mM K₂HPO₄, pH 9.5). Following incubation the leaves were transferred into distaining solution and incubated at room temperature with gentle shaking. Nuclei staining with DAPI was performed as previously described[65]. Briefly, hyphae were incubated for 10 min in the dark in PBS containing 1 μg/ml DAPI, the staining solution was removed and the sample was washed with PBS. For staining of vacuoles with CMAC, spores were germinated on cover slips, the cultures were incubated in GB medium with 10 μM CMAC for 20 min and then washed with fresh GB medium.

**Transformation of *A. thaliana***. Agrobacterium-mediated transformation of *A. thaliana* flowers was performed using the floral dip method[66]. Transgenic plants were selected on half-strength Murashige and Skoog medium with 0.5% sucrose, 0.8% agar and 2.5 mM MES at pH 5.7, containing 50 μg/ml of kanamycin. Three independent transgenic lines were generated with *bccrh1* overexpression and empty vector (control) plasmids. Protein expression was confirmed by western blot analysis (Supplementary Fig. 11). Homozygous T3 seeds were selected and used for all experiments.

**Yeast transformation, yeast two-hybrid assay, and yeast complementation assay**. The Matchmaker Gold Yeast Two-Hybrid (Y2H) System (Clontech, Palo Alto, CA, USA) was used to verify proteins interaction. The bait (pGBKT7) and prey (pGADT7) plasmids were co-transformed into yeast strain AH109 using the LiAc/SS carrier DNA/PEG method[67]. Yeast transformants were screened on selective dropout (SD)/-Trp-Leu medium to select yeast cells containing the desired plasmids (pGADT7 and pGBKT7). Positive protein interactions were assessed on SD/-Trp-Leu-His-Ade medium supplemented with X-α-galactosidase (X-α-gal) after being incubated at 28°C for four days. For the yeast complementation assay, sensitivity of the different yeast strains to Congo red was determined as described previously[28]. In brief, yeast cells were cultured in YPD or SD-Ura (for strains bearing YEp352 plasmid) medium at 24 °C overnight., The cells were collected by centrifugation at 5000 × g for 2 min, washed twice with ddH₂O, the cells were diluted to 3 × 10⁶ cells ml⁻¹ and then further five 1:5 serial dilutions. Cell suspension (five microliters) were spotted onto YPD medium and YPD supplemented with Congo red for 3 days at 30 °C.

**Reporting summary**. Further information on research design is available in the Nature Research Reporting Summary linked to this article.

## Data availability
The data supporting the results of this study are available within the article and its Supplementary Information files. The source data underlying Figs. 3a, 4a–f, 5a, 6b, f, 7b, d, f, and 8b, c and Supplementary Figs 5b, 6, 7, 9a–c, e, 10 and 11 are provided as a Source Data file. The authors declare that the other data supporting the findings of this study are available from the corresponding author upon request. Source data are provided with this paper.

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

## Acknowledgements

We are grateful to Prof. Adi Avni and Mrs. Orian Taig for provision of and assistance with tomato cell cultures. This research was supported by Research Grant Award No. IS-4937-16 from BARD, The United States - Israel Binational Agricultural Research and Development Fund to A.S. and grants BIO2016-79289-P and PID2019-105223GB-I00 (Ministerio de Economía y Competitividad, MINECO, Spain) to J.A.

## Author contributions

K.B., L.S., and A.S. designed the experiments, and K.B. and A.S. wrote the paper with inputs from all authors; K.B., L.S., W.Z., N.J., R.F., A.B.S., and W.Z. performed the experiments; T.M., J.A., and G.M. analyzed the data and provided critical feedback.

## Competing interests

The authors declare no competing interests.
