## [Peer Review File · Nature Communications]

REVIEWER COMMENTS

Reviewer #1 (Remarks to the Author):

The authors report the identification of BcCrh1 as a NIP protein that induces cell death and plant defense responses. These proteins has been previously characterized in other fungi as transglycosilases required for the crosslinking between cell wall chitin and glucan polymers. Interestingly, the authors propose a novel and unexpected function for this *Botrytis cinerea* protein in plant infection. Although there are interesting results in the manuscript, in my opinion there are also some important concerns that avoid its publication. Particularly, the authors conclude that dimerization of these proteins is necessary for fungal cell wall biosynthesis whereas monomers are involved in fungal plant interaction. However, they do not show solid evidences about this. Deletion or overexpression of BcCrh1 has no effect on fungal development or pathogenicity. However, overexpression of an inactive catalytic protein version rendered development defects and less virulent strains. This result is very interesting and the author's prediction from these experiments is that BcCrh1 should form dimers necessary for the transglycosylation activity. This reviewer do not see this argument. Moreover, no transglycosylation experiment (biochemical assay) is shown in agreement with this conclusion. Therefore, for me, the conclusion is completely overestimated. Chitin-glucan crosslinking biochemical assays has previously shown that Crh homologous in *Saccharomyces*, *Candida* and *Aspergillus* are able to use chitin as donor and beta 1,3 and beta-1,6 glucan as acceptors in a transglycosylation reaction. As part of this reaction it has also been shown that these proteins are able to act as chitinases. Therefore, an important author's claim is that dimerization is necessary for the transglycosylase activity but no biochemical evidences are shown.

The authors propose that these proteins form homodimers as deduced from two hybrid assays, but again no additional biochemical evidences are shown. These additional experiments are absolutely necessary. Moreover, since Cys are residues very important for protein stability, it is also necessary to demonstrate that Cys site directed mutant proteins are stably expressed and not degraded.

An additional concern is about the experiments shown in Figure 4. Apparently BcCrh1 also triggers plant defence responses. However I miss more explanations about the genes assayed in the panel b and comments about differences between 24 and 48h as well as between active and inactive proteins in these assays.

Reviewer #2 (Remarks to the Author):

The manuscript by Bi et al. describes the role the protein BcCrh1 in the interaction of the phytopathogenic fungus *botrytis cinerea* with plants. The experiments are all well conducted, the manuscript is well written, and the conclusions are sound. To my knowledge, this is the first example of a *B. cinerea* effector that enters plant cells and displays a cell-death promoting activity there, and I think this will be of broad interest. My advice is to accept the paper with only minor modifications.

Minor comments:

Although generally well written, the manuscript could benefit from a careful correction by a native English speaker.

The authors can try to make the figures more "appealing", especially in suppl. figures. In suppl. Fig 4, for example, labelling the lanes inside the picture does not give a good impression. And there is no need to do so, as there is not space imitation is supplementary figures. Alignment in suppl. Fig. 1 is also difficult to read, I do not see the point in such a reduction for a material designed to be published online and not in paper. Legend to suppl. Figures as misplaced.

L 41-43. The authors are reaching too far. This is unknown for most NIPs.

119-123. Although the experiment and the results are OK, the reasoning for the design of the experiment does not seem right, because the experiment in fig. 2 has not been explained yet. I would be better to explain experiment in fig 2 in full, and then introduce the hypothesis that region 21-74 is an uptake signal.

L 650-635. Please explain the meaning of M in MBcXYG1.

Fig. 5 What is OE.GFP?

Suppl. Fig. 8: "Notably, at this stage, all 72 strains except Δ /OE-MBcCrh1 formed normal infection

cushions." This is not apparent from the figure. The images are not particularly clear.

Fig. 8c. Is there an "A" missing in "C26AC33"?

Pag. 257 "The specific accumulation of BcCrh1 in infection cushions implies that it is necessary for infection cushion formation". This is pure speculation, and the fact that the mutant lacking BcCrh1 still forms infection cushions does not seem to corroborate it. You may substitute by " suggests that... contributes...".

Reviewer #3 (Remarks to the Author):

Comments and Suggestions for Authors

General: In this manuscript, the author revealed a novel and unexpected role of Crh proteins as mediators of fungal-plant interaction, and provides new details on their role in cell fungal wall biosynthesis. The aim of the research is interesting and of somewhat importance. Unfortunately, despite its potential value for the scientific community, the data presentation and evaluation in its present state requires editing.

Minor revision:

1. In this manuscript, many results are not solid enough and need additional experiments to further verify. Besides, some of the results were only images that lacked quantitative analysis. For example:

1) When analyzing necrotic lesions of leaves, the lesion should be stained with trypan blue and quantified.

2) When you want to check the PTI responses, callose deposition and H₂O₂ accumulation should be checked and quantified, such as calculating the average number of callose deposits per mm² and the DAB-stained area.

3) When you want to check the homo- and hetero-dimers of different BcCrh members, pull-down and coimmunoprecipitation assay (co-IP) should be used to further confirm the results in vitro and in planta.

4) When you want to check the lesion size of *B. cinerea* infected leaves, normally relative biomass of *B. cinerea* in the infected leaves also should be quantified.

5) If you want to verify co-localization, fluorescence intensity profile analysis is a better way.

2. There are some mistakes and questions in this manuscript, such as :

1) In line 18, "fugal" should be revised to "fungal".

2) In line 114, "Fig. 1b" should be revised to "Fig. 1b-g".

3) In line 155, "PTI genes" should be revised to "defense-related genes".

4) In the legend of Figure 4C, the concentration of infiltrated purified proteins should be noted.

5) In Figure 4b, significant difference analysis should be done.

6) In lines 174-175, the author mentioned that the protein was observed in the fungal vacuoles and ER, and accumulated to high levels in infection cushions. However, I can't find the ER marker. In Figure 5b, please re-check the scale bar. And I also recommend marking the infection cushions in the figures with red/white arrow.

7) In Figure 3b, why did the author use tomato not tobacco here?

8) In Figure 3c, why there is no GFP signal for GFP-BcCrh175-144?

9) I recommend that the author analyze characteristics of the two regions of amino acids which sufficient for protein uptake and compared with other fungi in the discussion part.

3. In this manuscript, some results are not described clearly, such as:

1) In line 108, the author mentioned that any of the four constructs induced similar levels of cell death? I just wonder that they are similar strong or similar weak of cell death?

2) In lines 104-105, the author mentioned that the native secretion signal was replaced with the secretion sequence of Arabidopsis pathogenesis-related protein 3 (PR3). The reason using secretion sequence of PR3 should also be described here.

3)

So, considering the quality of the writing is not good, I would suggest re-check the whole text by a native English speaker (I am not).

Overall, the story doesn't seem complete as the target of BcCrh1 hasn't been identified in planta and the reason why BcCrh1 did not promote necrosis in Arabidopsis also hasn't been well told.

Reviewer #1

Comment: The authors report the identification of BcCrh1 as a NIP protein that induces cell death and plant defense responses. These proteins has been previously characterized in other fungi as transglycosilases required for the crosslinking between cell wall chitin and glucan polymers. Interestingly, the authors propose a novel and unexpected function for this *Botrytis cinerea* protein in plant infection. Although there are interesting results in the manuscript, in my opinion there are also some important concerns that avoid its publication. Particularly, the authors conclude that dimerization of these proteins is necessary for fungal cell wall biosynthesis whereas monomers are involved in fungal plant interaction. However, they do not show solid evidences about this.

Deletion or overexpression of BcCrh1 has no effect on fungal development or pathogenicity. However, overexpression of an inactive catalytic protein version rendered development defects and less virulent strains. This result is very interesting and the author's prediction from these experiments is that BcCrh1 should form dimers necessary for the transglycosylation activity. This reviewer do not see this argument. Moreover, no transglycosylation experiment (biochemical assay) is shown in agreement with this conclusion. Therefore, for me, the conclusion is completely overestimated. Chitin-glucan crosslinking biochemical assays has previously shown that Crh homologous in *Saccharomyces*, *Candida* and *Aspergillus* are able to use chitin as donor and beta 1,3 and beta-1,6 glucan as acceptors in a transglycosylation reaction. As part of this reaction it has also been shown that these proteins are able to act as chitinases. Therefore, an important author's claim is that dimerization is necessary for the transglycosylase activity but no biochemical evidences are shown.

Response: To address this point we performed complementation experiments of a *Saccharomyces cerevisiae* $\Delta crh1/\Delta crh2$ double mutant. The mutant is completely deficient of Crh activity and therefore hypersensitive to Congo red (Crh stands for Congo red hypersensitivity). Expression of the *B. cinerea bccrh1* gene fully restored the wild type phenotype, showing that BcCrh1 has the necessary transglycosylase activity (Supplementary Figure 6). Furthermore, the BcCrh1 with mutations in the catalytic residues E120Q/D122H/E124Q (MBcCrh1) did not complement the yeast mutant, which confirms that this clone lost the enzymatic activity. Finally, we showed that the BcCrh1 with point mutations that prevent protein dimerization (BcCrh1^{C26AC33A}) also could not complement the yeast mutant (Supplementary Figure 6), which confirms that protein dimerization is essential for the enzymatic activity.

Comment: The authors propose that these proteins form homodimers as deduced from two hybrid assays, but again no additional biochemical evidences are shown. These additional experiments are absolutely necessary. Moreover, since Cys are residues very

important for protein stability, it is also necessary to demonstrate that Cys site directed mutant proteins are stably expressed and not degraded.

Response: To further address possible dimerization of BcCrh1 we performed co-immunoprecipitation in *N. benthamiana*, *in vitro* pull down and split GFP (BiFC) experiments (see Figure. 8). All of these assays show clear interaction of the BcCrh1 monomers.

Stability of the Cys mutant protein was evaluated by immunoblot analysis of the targeted proteins, including native protein BcCrh1 and Cys site directed mutant protein BcCrh1^{C26AC33A}. The results show good expression of both peptides, with same molecular weight (Supplementary Fig. 6, Supplementary Fig. 11). Additionally, the BcCrh1^{C26AC33A} mutant protein retained full necrosis-inducing activity as demonstrated by agro-infiltration assay (Supplementary Fig. 3b, Supplementary Fig. 5). These results confirm that the Cys site directed mutant peptide BcCrh1^{C26AC33A} is stably expressed and retains the necrosis-inducing activity.

Comment: An additional concern is about the experiments shown in Figure 4. Apparently BcCrh1 also triggers plant defence responses. However I miss more explanations about the genes assayed in the panel b and comments about differences between 24 and 48h as well as between active and inactive proteins in these assays.

Response: The eight defense genes that we used to monitor activation of the plant defense were selected according to published studies as representative of selected defense pathways:

PR1a: A marker gene for activation of defense responses and the “gold standard” of SAR (systemic acquired resistance).

NPR1: A key regulator of the salicylic acid (SA)-mediated SAR pathway.

ACS6: Ethylene biosynthesis and responsive marker gene.

OPR3: Jasmonic acid (JA) biosynthesis.

PI-II: One of the prominent genes in wounding and associated signaling molecules.

GluB: Encodes for Glucan endo-1,3-beta-glucosidase, an important plant defense-related product against fungal pathogens and an important marker gene in plant defense.

LRR22: The tomato *LRR22* gene has been used as a marker for PAMP-triggered immunity.

LoxD: A marker gene for systemin-induced gene expression.

To test effect of the protein on gene expression, tomato leaves were infiltrated with GFP, BcCrh1, or MBcCrh1 protein solutions (11 µM), total RNA was extracted at 0, 24 and 48 h after protein infiltration, followed qRT-PCR analysis for measurement of the plant defense-related marker gene expression levels. In the revised version (Figure 4c), we rearranged the histograms by different time points (0, 24 and 48 h) for better interpretation. According to our result, the expression levels of most selected plant

defense-related marker genes were induced at both 24 h and 48 h after protein infiltration (except for *LoxD* that was induced only at 24 h), and some of the genes (*GluB*, *PR1a*, *PI-II*) showed higher expression levels at 48 h than 24 h, indicating gradual induction of these genes.

The gene expression data imply that the inactive protein (MBcCrh1) triggers slightly stronger activation of defense responses than the native protein. While the mechanism is yet to be revealed, this result hints that it will be better to use the MBcCrh1 for the engineering pathogen-resistance in plants.

Reviewer #2

The manuscript by Bi et al. describes the role of the protein BcCrh1 in the interaction of the phytopathogenic fungus *Botrytis cinerea* with plants. The experiments are all well conducted, the manuscript is well written, and the conclusions are sound. To my knowledge, this is the first example of a *B. cinerea* effector that enters plant cells and displays a cell-death promoting activity there, and I think this will be of broad interest. My advice is to accept the paper with only minor modifications.

Minor comments:

Comment: Although generally well written, the manuscript could benefit from a careful correction by a native English speaker.

Response: The revised version of the manuscript was edited by professional language editor.

Comment: The authors can try to make the figures more “appealing”, especially in supplementary figures. In suppl. Fig 4, for example, labelling the lanes inside the picture does not give a good impression. And there is no need to do so, as there is not space imitation in supplementary figures. Alignment in suppl. Fig. 1 is also difficult to read, I do not see the point in such a reduction for a material designed to be published online and not in paper. Legend to suppl. Figures as misplaced

Response: Thank you for these suggestions. We have removed labelling from Supplementary Fig. 11 (revised version), as well as modified other figures accordingly. Supplementary Fig. 1 was reformatted (mainly enlarged) for better clarity. The legend to suppl figures was corrected.

L 41-43. The authors are reaching too far. This is unknown for most NIPs.

Response: The text was modified to avoid overstatement (L43-47 in revised version).

119-123. Although the experiment and the results are OK, the reasoning for the design of the experiment does not seem right, because the experiment in fig. 2 has not been

explained yet. It would be better to explain experiment in fig 2 in full, and then introduce the hypothesis that region 21-74 is an uptake signal.

Response: The text was modified as suggested.

L 650-653. Please explain the meaning of M in MBcXYG1.

Response: The M stands for mutations at the active site residues which renders the BcXYG1 enzyme inactive. Text was added in the revised Figure 3 legend (L703-704).

Fig. 5 What is OE.GFP?

Response: OE.GFP is the strain that over expresses free GFP protein. The information was added in the description of *B. cinerea* strains used in this study (Supplemental Table 2).

Suppl. Fig. 8: “Notably, at this stage, all strains except Δ /OE-MBcCrh1 formed normal infection cushions.” This is not apparent from the figure. The images are not particularly clear.

Response: We have replaced the original images with a better quality images. The typical and abnormal infection cushions are indicated differently (Supplementary Fig. 9 in revised version).

Fig. 8c. Is there an “A” missing in “C26AC33”?

Response: Thanks for noticing. Corrected.

Pag. 257 “The specific accumulation of BcCrh1 in infection cushions implies that it is necessary for infection cushion formation”. This is pure speculation, and the fact that the mutant lacking BcCrh1 still forms infection cushions does not seem to corroborate it. You may substitute by “ “suggests that... contributes...””.

Response: Good point. Sentence was modified as suggested (L268-271).

Reviewer #3

Comments and Suggestions for Authors

General: In this manuscript, the author revealed a novel and unexpected role of Crh proteins as mediators of fungal-plant interaction, and provides new details on their role in cell fungal wall biosynthesis. The aim of the research is interesting and of somewhat importance. Unfortunately, despite its potential value for the scientific community, the data presentation and evaluation in its present state requires editing.

Minor revision:

1. In this manuscript, many results are not solid enough and need additional experiments to further verify. Besides, some of the results were only images that lacked quantitative analysis. For example:

Response: Thanks for these suggestions. We performed additional experiments as well as quantitative and statistical analyses to improve data quality. Please see description of changes in the specific responses below.

1) When analyzing necrotic lesions of leaves, the lesion should be stained with trypan blue and quantified.

Response: We have conducted new Agro-infiltration assays and stained the leaves with trypan blue. Lesion size was quantified by measurement of trypan blue staining intensity using ImageJ software (Supplementary Figure 5).

2) When you want to check the PTI responses, callose deposition and H₂O₂ accumulation should be checked and quantified, such as calculating the average number of callose deposits per mm² and the DAB-stained area.

Response: We performed the suggested assays of PTI responses: quantification of ROS callose deposition following staining with DAB and aniline blue, respectively (Figure 4a and b).

3) When you want to check the homo- and hetero-dimers of different BcCrh members, pull-down and coimmunoprecipitation assay (co-IP) should be used to further confirm the results *in vitro* and *in planta*.

Response: We have verified the BcCrh1 dimerization by three additional methods: Co-IP in *N. benthamiana* plants, BiFC assay in *Botrytis*, and *in vitro* Pull-Down assay (see Figure 8).

4) When you want to check the lesion size of *B. cinerea* infected leaves, normally relative biomass of *B. cinerea* in the infected leaves also should be quantified.

Response: Measurement of fungal biomass in infected tissues is a common practice in many studies, and is particularly relevant in case of biotrophic or hemibiotrophic pathogens, in which symptoms are not directly correlated with fungal biomass. For *B. cinerea*, it is less important since lesion size is completely correlated with biomass, therefore it is used mainly for quantification of differences between treatments with a relatively small effect on virulence. In our study, infection assays either show no effect, or a substantial effect (by over expression of MBcCrh1), and we don't aim at quantification of the differences since it has no relevance to the objectives of the study. Therefore, we feel that in this case measurement of biomass will have no added value over measurement of lesion diameter.

5) If you want to verify co-localization, fluorescence intensity profile analysis is a better way.

Response: We added fluorescence intensity profile analysis to verify the co-localization (Figure. 5a).

2. There are some mistakes and questions in this manuscript, such as :

1) In line 18, “fugal” should be revised to “fungal”.

Response: Corrected (L20).

2) In line 114, “Fig. 1b” should be revised to “Fig. 1b-g”.

Response: Thanks for pointing out this problem! Corrected as suggested.

3) In line 155, “PTI genes” should be revised to “defense-related genes”.

Response: Corrected (L163).

4) In the legend of Figure 4C, the concentration of infiltrated purified proteins should be noted.

Response: Values of infiltrated purified proteins were added to the Figure legend.

5) In Figure 4b, significant difference analysis should be done.

Response: Statistical differences analysis at $P \leq 0.01$ using one-way ANOVA was included in the revised version (Figure 4c).

6) In lines 174-175, the author mentioned that the protein was observed in the fungal vacuoles and ER, and accumulated to high levels in infection cushions. However, I can't find the ER marker. In Figure 5b, please re-check the scale bar. And I also recommend marking the infection cushions in the figures with red/white arrow

Response: The localization to ER is based on the observation of the signal around nuclei. Fungal nuclei were stained with DAPI, then the samples were visualized using a Confocal microscope. The GFP signal is observed around nuclei, which is indicative of ER localization, and in vacuoles as indicated by overlap of the GFP signal with the vacuole dye. We checked the scale bar in Figure 5b and confirmed that it is correct. We added arrows to denote infection cushions.

7) In Figure 3b, why did the author use tomato not tobacco here?

Response: Based on the result from response of tomato leaves and *N. benthamiana* to purified BcCrh1 protein, we found that tomato leaves are more susceptible to BcCrh1 than *N. benthamiana* leaves (Supplementary Fig. 4). In addition, tomato cell cultures

(Msk8) were available to us, which were used for translocation experiments with a GFP-tagged peptide.

8) In Figure 3c, why there is no GFP signal for GFP-BcCrh1⁷⁵⁻¹⁴⁴?

Response: For the uptake assay of GFP-labeled proteins by plant cells, tomato cell cultures (Msk8) were incubated with GFP-tagged peptide solutions for 18 h. To eliminate background of GFP signal that was not taken into the cells, the cells were subsequently washed three times in incubation buffer (10 mM phosphate buffer, pH 7, and 0.1% BSA) and then visualized by aconfocal microscope. Without the functional uptake signal BcCrh1²¹⁻⁷⁴, GFP-BcCrh1⁷⁵⁻¹⁴⁴ couldn't be delivered into the plant cells, therefore after washing of the incubation buffer, no GFP signal was detected in cells that were subjected to this treatment.

9) I recommend that the author analyze characteristics of the two regions of amino acids which sufficient for protein uptake and compared with other fungi in the discussion part.

Response: We added discussion of this point in the revised text (L280-291).

3. In this manuscript, some results are not described clearly, such as:

1) In line 108, the author mentioned that any of the four constructs induced similar levels of cell death? I just wonder that they are similar strong or similar weak of cell death?

Response: We used trypan blue stain and quantified the result accordingly. Accord to the quantified data, the four constructs induced similarly strong levels of cell death. We have rephrased the sentence accordingly (L114-115).

2) In lines 104-105, the author mentioned that the native secretion signal was replaced with the secretion sequence of Arabidopsis pathogenesis-related protein 3 (PR3). The reason using secretion sequence of PR3 should also be described here.

Response: We have added a citation for explanation the reason using secretion sequence of PR3 from *Arabidopsis* in the revised text (L112).

3) So, considering the quality of the writing is not good, I would suggest re-check the whole text by a native English speaker (I am not).

Response: The manuscript was edited by a professional language editor.

Comment: Overall, the story doesn't seem complete as the target of BcCrh1 hasn't been identified in planta and the reason why BcCrh1 did not promote necrosis in *Arabidopsis* also hasn't been well told.

Response: BcCrh1 is a cytoplasmic effector, unlike all previously reported catalytic NIPs, which remain in the plant apoplastic space. Our current research indicated that the interaction site of BcCrh1 with its putative plant target is in the plant cytosol space, since a nuclear localized BcCrh1 was inactive (Supplementary Fig. 3c). We believe that this preliminary result will inspire further screening and identification of the BcCrh1 target in plants, which will require a new study. More research will also be necessary to unravel the mechanistic of necrosis induction, which may provide an answer for the question why BcCrh1 did not promote necrosis in *Arabidopsis*.

REVIEWERS' COMMENTS

Reviewer #2 (Remarks to the Author):

After carefully reviewing the manuscript for a second time, and in view of the changes introduced by the authors, I think that all the points raised by this reviewer have been addressed adequately. I have no further comments.

Reviewer #3 (Remarks to the Author):

[No further comments for author]

Reviewer #4 (Remarks to the Author):

I have been asked by the editor to comment on the response of the authors to the first two comments of reviewer 1, concerning the need for dimerization of BcCrh1 for transglycosylation activity (and, thus, presumably also for chitinase activity), but not for necrosis inducing activity in the plant cell. In response to these comments, the authors have included a complementation assay in which the Botrytis gene, but neither a presumably catalytically inactivated mutant nor a cys-ala mutant which is assumed to no longer dimerise (see below), is complementing the Congo red hypersensitivity phenotype of a *S. cerevisiae* crh knockout (Supplementary Figure 6). I agree with the conclusion of the authors that this tends to indicate that BcCrh1 has similar enzyme activity as Crh from *S. cerevisiae*. However, as the reviewer already stated, inability of the cys-ala mutant to compensate for the lack of Crh activity does not necessarily imply that dimerization is required for enzyme activity; instead, the mutation might lead to misfolding, leading to enzymatic inactivity or, if misfolding is pronounced, also to degradation. Using SDS-PAGE, the authors show that the enzyme is not degraded, but they do not show that it is correctly folded. This is still a possibility.

The authors have also added co-IP, pull-down, and split-GFP experiments to show interaction of the BcCrh1 monomers, but not of the cys-ala mutant (Figure 8). However, co-IP and pull-down were apparently not performed using the cys-ala mutant, leaving only the split-GFP experiments for the conclusion that this mutant does not dimerize any more. Again, I tend to agree that the results show dimerization (but the legend of Fig. 8 b and c needs improvement!), and also that the cys-ala mutant does not dimerize. However, it does not prove that dimerization is required for enzyme activity. Perhaps, the misfolding of the cys-ala mutant has two independent consequences: loss of enzyme activity and loss of dimerization. That dimerization is required for enzyme activity cannot be concluded safely from these results.

In my opinion, the paper tells two stories, namely (1) that in addition to its expected enzyme activity as a chitin-glucan crosslinking transglycosylase, BcCrh1 has an unexpected necrosis inducing activity in plant cells that is independent of its enzymatic activity; and (2) that BcCrh1 can dimerise and requires dimerization for enzymatic activity. In my opinion, conclusion (1) is justified, while conclusion (2) is not. Clearly, both stories would win if direct biochemical evidence for (chitinase and) transglycosidase activity would be given, and for lack of activity in the catalytic and cys-ala mutants. A biochemical activity assay could also clearly show whether the monomer is enzymatically active or whether dimerization is required for enzyme activity.

Bruno Moerschbacher

Reviewer #4 (Remarks to the Author):

Comment: I have been asked by the editor to comment on the response of the authors to the first two comments of reviewer 1, concerning the need for dimerization of BcCrh1 for transglycosylase activity (and, thus, presumably also for chitinase activity), but not for necrosis inducing activity in the plant cell. In response to these comments, the authors have included a complementation assay in which the Botrytis gene, but neither a presumably catalytically inactivated mutant nor a cys-ala mutant which is assumed to no longer dimerise (see below), is complementing the Congo red hypersensitivity phenotype of a *S. cerevisiae* crh knockout (Supplementary Figure 6). I agree with the conclusion of the authors that this tends to indicate that BcCrh1 has similar enzyme activity as Crh from *S. cerevisiae*. However, as the reviewer already stated, inability of the cys-ala mutant to compensate for the lack of Crh activity does not necessarily imply that dimerization is required for enzyme activity; instead, the mutation might lead to misfolding, leading to enzymatic inactivity or, if misfolding is pronounced, also to degradation. Using SDS-PAGE, the authors show that the enzyme is not degraded, but they do not show that it is correctly folded. This is still a possibility.

Response: The possibility of dimerization was deduced from the phenotype of the BcCrh1 fungal mutants: deletion or over expression had no clear effect on the fungus, but over expression of the inactive form of BcCrh1, and even more so, a double mutant with deletion of the native copy and over-expression of the enzyme inactive form of BcCrh1, had clear developmental defects (figure 6). Based on these results we hypothesized the dimerization model, and accordingly predicted that over production of the BcCrh1 inactive form would titer the active monomers (of either BcCrh1 as well as BcCrh2 and BcCrh3, see figure 8 and proposed model above). This model is supported by two additional strains: a strain that expresses the cys-ala BcCrh1 mutant from the strong Olic promoter, and a strain that expresses the enzyme inactive from the much weaker, native BcCrh1 promoter, both in background of *bccrh1* deletion. Both strains did not show any of the developmental defects associated with over expression (Olic promoter) of the enzyme in active form (Figure 7). Furthermore, deletion of *BcCrh1* or over expression of the native form have no clear phenotypes. Collectively, these results rule out the possibility that loss of activity of the monomeric peptide due to the cy-ala mutation accounts to the observed phenotypes, and can only be explained by some type of interactions between the BcCrh monomers. The yeast complementation results do not stand alone but rather substantiate this possibility.

Comment: The authors have also added co-IP, pull-down, and split-GFP experiments to show interaction of the BcCrh1 monomers, but not of the cys-ala mutant (Figure 8). However, co-IP and pull-down were apparently not performed using the cys-ala mutant, leaving only the split-GFP experiments for the conclusion that this mutant does not dimerize anymore. Again, I tend to agree that the results show dimerization (but the legend of Fig. 8 b and c needs improvement!), and also that the cys-ala mutant does not dimerize. However, it does not prove that dimerization is required for enzyme activity. Perhaps, the misfolding of the cys-ala mutant has two independent consequences: loss of enzyme activity and loss

of dimerization. That dimerization is required for enzyme activity cannot be concluded safely from these results.

Response: First, thanks for comment on Figure 8 legend; the legend has been corrected. As far as the connection between dimerization and activity, as explained above, the complementation assay should be considered together with the other relevant results, namely mutant phenotypes and hybridization. We agree with the comment that the mutations could theoretically lead to miss folding and loss of activity of the protein, although based on the 3d structure and the position of the two cysteine residues (Suppl Figure 2) it is unlikely that the mutation significantly affected protein folding. However, even if it did, loss of activity due to the cys-ala mutations can't explain the phenotype of the different fungal mutants. We are convinced that collectively, the results from the different lines of work (mutant phenotypes, hybridization, yeast complementation assay) provide a sound proof for the connection between transglycosylases activity and protein dimerization.

Comment: In my opinion, the paper tells two stories, namely (1) that in addition to its expected enzyme activity as a chitin-glucan crosslinking transglycosylase, BcCrh1 has an unexpected necrosis inducing activity in plant cells that is independent of its enzymatic activity; and (2) that BcCrh1 can dimerise and requires dimerization for enzymatic activity. In my opinion, conclusion (1) is justified, while conclusion (2) is not. Clearly, both stories would win if direct biochemical evidence for (chitinase and) transglycosidase activity would be given, and for lack of activity in the catalytic and cys-ala mutants. A biochemical activity assay could also clearly show whether the monomer is enzymatically active or whether dimerization is required for enzyme activity.

Response: We agree with the reviewer that a direct biochemical chitinase and transglycosidase activity assay would be helpful. To be on the safe side, we toned down the conclusion about dimerization/transglycosylases activity throughout the manuscript.